# Multi-omic and functional analysis for classification and treatment of sarcomas with FUS-TFCP2 or EWSR1-TFCP2 fusions

Julia Schöpf[1,2,3,27], Sebastian Uhrig[4,5,27], Christoph E. Heilig [2,5,27], Kwang-Seok Lee [2,27], Tatjana Walther[2], Alexander Carazzato [2], Anna Maria Dobberkau[6], Dieter Weichenhan[7], Christoph Plass[7], Mark Hartmann [6], Gaurav D. Diwan[8,9], Zunamys I. Carrero [10,11], Claudia R. Ball[10,11,12,13], Tobias Hohl [1,3], Thomas Kindler[14,15,16], Patricia Rudolph-Hähnel[14,15,16], Dominic Helm [17], Martin Schneider[17], Anna Nilsson[18], Ingrid Øra [19], Roland Imle [20,21,22], Ana Banito [20,21], Robert B. Russell [8,9], Barbara C. Jones[5,21,22], Daniel B. Lipka [6], Hanno Glimm [10,11,12,23], Daniel Hübschmann [4,5,24], Wolfgang Hartmann [25], Stefan Fröhling [2,5,26,28] ✉ & Claudia Scholl [1,28] ✉

Linking clinical multi-omics with mechanistic studies may improve the understanding of rare cancers. We leverage two precision oncology programs to investigate rhabdomyosarcoma with FUS/EWSR1-TFCP2 fusions, an orphan malignancy without effective therapies. All tumors exhibit outlier ALK expression, partly accompanied by intragenic deletions and aberrant splicing resulting in ALK variants that are oncogenic and sensitive to ALK inhibitors. Additionally, recurrent *CKDN2A/MTAP* co-deletions provide a rationale for PRMT5-targeted therapies. Functional studies show that FUS-TFCP2 blocks myogenic differentiation, induces transcription of ALK and truncated TERT, and inhibits DNA repair. Unlike other fusion-driven sarcomas, TFCP2-rearranged tumors exhibit genomic instability and signs of defective homologous recombination. DNA methylation profiling demonstrates a close relationship with undifferentiated sarcomas. In two patients, sarcoma was preceded by benign lesions carrying FUS-TFCP2, indicating stepwise sarcomagenesis. This study illustrates the potential of linking precision oncology with preclinical research to gain insight into the classification, pathogenesis, and therapeutic vulnerabilities of rare cancers.

Soft-tissue sarcomas (STS) are mesenchymal malignancies that are characterized by histologic diversity and a variable clinical course[1]. Rhabdomyosarcoma (RMS) is an STS subtype composed of immature precursor cells with partial myogenic differentiation defined by aberrant expression of the myogenic transcription factors MYOD1 and MYOG[2,3]. It is the most common form of STS in children but can arise at any age[2,4]. The current World Health Organization (WHO) classification distinguishes embryonal (ERMS), alveolar (ARMS), pleomorphic (PRMS), and spindle cell/sclerosing RMS based on histologic, genetic, and clinical features[5]. ERMS are genetically heterogeneous and exhibit DNA copy-number alterations and RAS pathway mutations. ARMS are characterized by PAX3/PAX7-FOXO1 fusions[5], and PRMS display

complex karyotypes and TP53 inactivation[2,3]. Spindle cell/sclerosing RMS may carry gene fusions involving, e.g., VGLL2 and NCOA2 or MYOD1 L122R mutations.

Beyond these subtypes, the WHO classification mentions intraosseous spindle cell RMS, which can carry MEIS1-NCOA2, FUS-TFCP2, or EWSR1-TFCP2 fusions[5]. Fusions of TFCP2 to either FUS or EWSR1 were recently discovered in three RMS cases[6], followed by additional reports identifying them in atypical or unclassified RMS[7–12]. This eventually led to the inclusion of TFCP2-rearranged RMS in the updated WHO classification; however, the molecular pathogenesis and clinically actionable vulnerabilities of these tumors are unknown.

TFCP2, together with TFCP2L1 and UBP1, belongs to one of two subfamilies of TFCP2/Grainyhead transcription factors, while the other comprises the three Grainyhead-like factors GRHL1-3[13,14]. TFCP2 has been linked to Alzheimer's disease and the expression of HIV genes and acts as an oncogene in hepatocellular carcinoma, pancreatic ductal adenocarcinoma, and breast cancer but exerts a tumor-suppressive function in melanoma[13]. The only known genetic alterations affecting TFCP2 in cancer are the recently discovered fusions to FUS and EWSR1 in atypical RMS. FUS and EWSR1, members of the FET protein family involved in many cellular processes[15], have similar structures[16] and are involved in transcriptional regulation, RNA splicing, and DNA damage repair[15,17–19]. Furthermore, they constitute the 5' part of chimeric fusions with different transcription factors that drive various sarcomas and leukemias[20,21].

In this study, we characterize TFCP2-rearranged RMS in multiple dimensions. We evaluate the histopathologic and clinical features of 12 cases and determine their genomic, transcriptomic, and epigenomic landscapes compared with those of 1,310 patients enrolled in a pan-cancer precision oncology program[22]. Furthermore, we investigate the oncogenic properties of the underlying FUS/EWSR1-TFCP2 fusions and alterations of the ALK receptor tyrosine kinase (RTK), which invariably accompany these rearrangements, and determine the fusions' direct and indirect transcriptional effects contributing to sarcoma development. Our data indicate that FUS/EWSR1-TFCP2-positive sarcoma is a distinct disease entity that does not fit the pattern of genetically silent fusion-driven sarcomas and provide insight into potential therapeutic vulnerabilities that may inform clinical decision-making in this difficult-to-treat sarcoma subtype.

## Results

### Clinical and histopathologic characteristics of FUS/EWSR1-TFCP2 RMS

We identified 12 patients with FUS-TFCP2 ($n = 8$) or EWSR1-TFCP2 ($n = 4$) in the prospective precision oncology programs MASTER[22] (Molecularly Aided Stratification for Tumor Eradication Research; adults, $n = 9$) and INFORM[23] (Individualized Therapy for Relapsed Malignancies in Childhood; children and adolescents, $n = 3$) (Supplementary Data 1). The median age at diagnosis was 36.5 years (range, 9–60), and most patients had localized disease ($n = 11$), with the primary tumor preferentially occurring in the head and neck region ($n = 9$) and bone ($n = 8$). Distant metastases occurred in six of the 11 patients initially presenting with localized disease, mainly in the lung and lymph nodes.

Primary treatments were heterogeneous, ranging from surgery alone to multimodal treatments including perioperative polychemo- and radiotherapy. All patients relapsed, mostly locoregionally, at a median of 6.5 months after treatment initiation, and the median overall survival was 29 months (range, 9–48). Four of the eight patients treated with neoadjuvant chemotherapy experienced disease progression during first-line treatment. The remaining four patients who completed chemotherapy and surgery had longer progression-free survival than patients treated with local modalities only (median, 13 months [range, 9–28] versus 6.5 months [range, 4–7]). Partial responses were observed in one patient who received neoadjuvant

ifosfamide, vincristine, doxorubicin, and actinomycin (TFCP2-HD-12) and one patient treated with doxorubicin and ifosfamide combined with regional hyperthermia with palliative intent (TFCP2-HD-10). In total, 34 regimens were given (median per patient, $n = 2$; range, 1–5), resulting in a median progression-free survival of two months (range, 1–8). The best responses reported were mixed response, stable disease, and progressive disease with one, nine, and 16 regimens applied, respectively. Thus, conventional chemotherapy showed no significant activity in our cohort of FUS/EWSR1-TFCP2 RMS (Supplementary Data 1).

Six patients received ALK inhibitors in later treatment lines, with radiographic response data available in three, and one patient received an ALK inhibitor as maintenance therapy after polychemotherapy. In one patient (TFCP2-HD-2) with lung metastases who received crizotinib as a third-line treatment, some metastases responded while most disease manifestations progressed. The other two patients showed no response to crizotinib (TFCP2-HD-1; fourth line) or alectinib (TFCP2-HD-6; fourth line) (Supplementary Data 1).

Histopathology revealed a morphologic spectrum of FUS/EWSR1-TFCP2 RMS. This comprised lesions with loose dermal and subcutaneous infiltrates of spindle cells with few intermingled epithelioid/rhabdoid cells, the latter only evident in desmin stains (pattern A; Supplementary Fig. 1a); lesions with hybrid spindle and epithelioid/rhabdoid morphology, consisting of dense, plump spindle cells with moderate amounts of eosinophilic cytoplasm arranged in short fascicles with a quantitatively heterogeneous component of fused epithelioid/rhabdoid cells (pattern B; Supplementary Fig. 1b); and lesions with hybrid spindle and epithelioid/rhabdoid morphology, including areas of considerable pleomorphism and/or blue cell/rhabdoid morphology (pattern C; Supplementary Fig. 1c, d; Supplementary Data 1). In most cases, nuclei had an ovoid shape with variable hyperchromasia; however, considerable nuclear irregularities were discernible in individual cases, particularly in recurrent lesions. Mitotic activity varied in these lesions, ranging from 0–72/10 high power fields (HPF), and occasional tumor necrosis was observed. In the tumor samples available for immunohistochemistry, heterogeneous desmin positivity (12/13), inconstant MYOD1 (12/13) and myogenin (9/13) expression, and at least partial ALK (9/13) positivity were observed (Fig. 1a, b and Supplementary Fig. 1).

Two patients (TFCP2-HD-4 and HD-5) had tissue from a primary lesion diagnosed as a benign tumor and a malignant recurrence occurring in the same location. In TFCP2-HD-5, the cellularity of the primary lesion was low, with bland, isomorphic spindle cells infiltrating the dermis and subcutaneous fat, and no mitotic figures were detected in 10 HPF. The corresponding recurrence had markedly increased cellularity and consisted mainly of spindle cells with intermingled epithelioid/rhabdoid elements, nuclear atypia was evident, and 23 mitoses were detected in 10 HPF. The immunohistochemical profile was comparable in both lesions (Fig. 1a). In TFCP2-HD-4, cellularity in the primary lesion was low but slightly increased compared to TFCP2-HD-5, showing infiltrative growth of spindle cells with minimal to low nuclear atypia in the dermis and subcutaneous fat, and no mitotic figures were detected in 10 HPF. A biphasic pattern was observed in the recurrent tumor: Larger areas resembled the spindle cell proliferation of the primary tumor, but cellularity was markedly increased, nuclear irregularities were readily apparent, and the mitotic count was 2/10 HPF. In addition, a sharply demarcated nodular tumor component was found, displaying a blue cell aspect and consisting of epithelioid/rhabdoid cells with coarse chromatin and 72 mitotic figures in 10 HPF. Immunohistochemically, the cellular component showed substantially reduced expression of desmin and myogenin but strong ALK positivity (Fig. 1b).

In summary, FUS/EWSR1-TFCP2 RMS is a morphologically variable, highly aggressive disease that seems to evolve from precursor lesions, is refractory to standard therapies, and has, at advanced stages following extensive pretreatment, limited sensitivity to ALK inhibition.

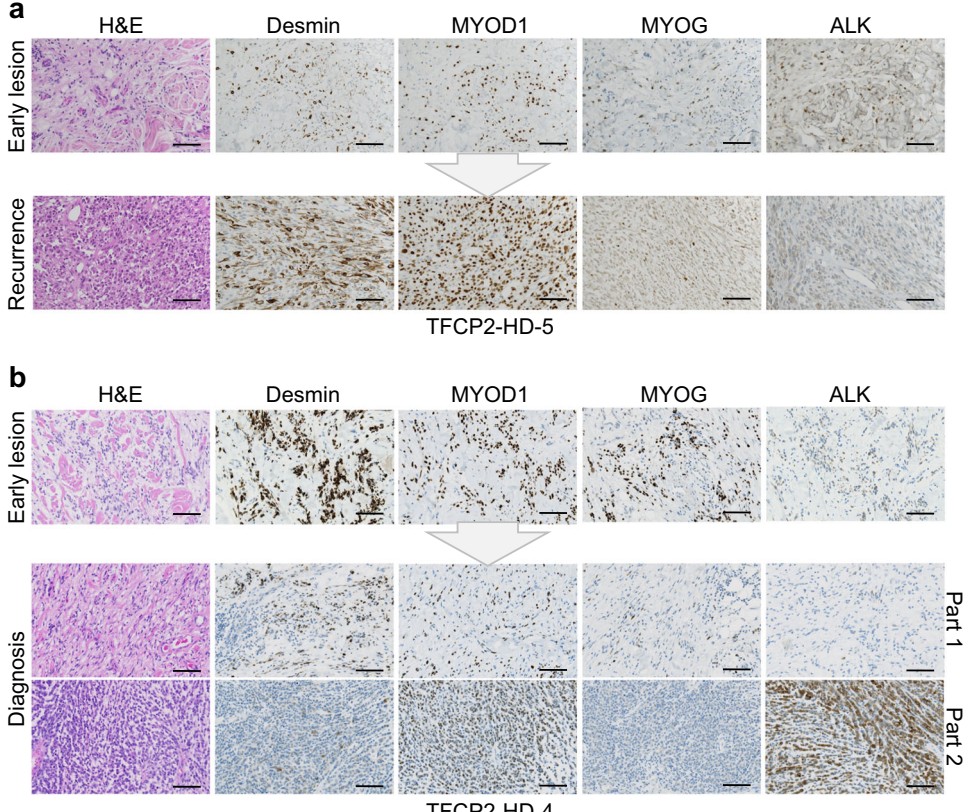

**Fig. 1 | Morphologic and immunohistochemical characteristics of FUS/EWSR1-TFCP2 RMS. a** Early lesion (top) showing isomorphic bland spindle cells infiltrating the dermis and subcutaneous fat. Recurrence with histopathologic criteria of malignancy (bottom) 32 months later with significantly increased cellularity, nuclear atypia, and brisk mitoses, composed mainly of spindle cells with intermingled epithelioid/rhabdoid elements. **b** Early lesion (top) with slightly increased cellularity compared to (a) showing infiltration of the dermis and subcutaneous fat by spindle cells with minimal to low nuclear atypia. Recurrent lesion (bottom) at RMS diagnosis 12 months later with a biphasic pattern consisting of larger areas resembling the spindle cell proliferation of the primary tumor with markedly increased cellularity and nuclear irregularities (part 1) and a second component with a blue cell aspect due to epithelioid/rhabdoid cells with atypical nuclei, coarse chromatin, and brisk mitotic activity (part 2). Stainings were performed once in an accredited pathology laboratory with standardized semi-automated procedures and appropriate controls. H&E, hematoxylin and eosin. Scale bar, 100 μm.

## Molecular landscape of FUS/EWSR1-TFCP2 sarcoma

FUS-TFCP2 and EWSR1-TFCP2 consisted of the first 254 and 138 N-terminal amino acids (aa) of FUS and EWSR1, respectively, and 460 aa of the TFCP2 protein. This corresponded to the fusion of the transcriptional activation domain and part of the first RGG1 domain of FUS or part of the transcriptional activation domain of EWSR1 with all functional domains of TFCP2 (Supplementary Fig. 2a).

Analysis of RNA sequencing (RNA-seq) data of 985 tumor samples from 967 patients enrolled in the MASTER program[22] and 15 samples of the 12 TFCP2-rearranged cases showed that the latter stood out by extremely high ALK expression (Fig. 2a). Closer examination of the *ALK* locus revealed different in-frame intragenic deletions (del) in five patients, i.e., del(ex1–16), del(ex2–16), del(ex3–16), del(ex1), del(ex2–3), and del(ex6–11) (Supplementary Fig. 2b and Supplementary Data 1). These led to the expression of short transcripts (ST), which we named ALK-ST1 (consisting of exons 1–2:18–29), ALK-ST2 (1:18–29), ALK-ST3 (18–29), and ALK-ST4 (1–5:12–17), through the loss of exons and additional splicing events, particularly skipping of exon 17 (Fig. 2b; Supplementary Fig. 2c and Supplementary Data 1). Furthermore, we detected the expression of three exons upstream of ALK (exons −3, −2, and −1), of which −2 and −1 replaced the regular exon 1 in the ALK transcript in eight tumors, suggesting control of ALK expression by an undescribed alternative promoter (Fig. 2b and Supplementary Fig. 2c). Together, these alterations resulted in truncated ALK proteins that lack different parts of the extracellular domain while retaining the transmembrane and kinase domains (ALK-ST1, -ST2, and -ST3) or lack all but certain portions of the extracellular domain (ALK-ST4; Supplementary Fig. 2d). ALK-ST3 is similar to the previously described oncogenic transcript variant initiating from an alternative transcription initiation (ATI) site in *ALK* intron 19 found in melanoma and STS[24,25], except that ALK-ST3 still contains the transmembrane domain (Supplementary Fig. 2d).

Unusual for fusion-driven sarcomas, whole-exome (WES) or whole-genome sequencing (WGS) revealed highly rearranged genomes in several cases, including chromothripsis in two tumors (Supplementary Fig. 2e). The tumors' genomic instability could be explained by homologous recombination deficiency (HRD), as evidenced by elevated HRD-LOH (loss of heterozygosity)[26], HRD-LST (large-scale state transition)[27], and HRD-TAI (telomeric allelic imbalance)[28] scores. A previous integrative analysis of these scores[29] showed that HR-deficient tumors had HRD-LOH, -LST, and -TAI values > 16.5, >10.2, and >13.7, respectively, or an average of these scores >13.4. At least one of these values was exceeded in six of the 12 FUS/EWSR1-TFCP2 tumors (Supplementary Fig. 2e).

Additional recurrently altered genes included *CDKN2A*, which was affected by copy-number loss in eight cases, and a frameshift deletion in one, resulting in a frequency of CDKN2A inactivation of 75% (Fig. 2c, d). *CDKN2A* was always co-deleted with the neighboring *MTAP* gene, whose loss confers sensitivity to PRMT5 inhibition[30,31]. Mechanistically similar to the loss of *CDKN2A*, five TFCP2-rearranged tumors were among the top 10% expressors of CCND2 in the MASTER cohort, two of which had a genomic gain of

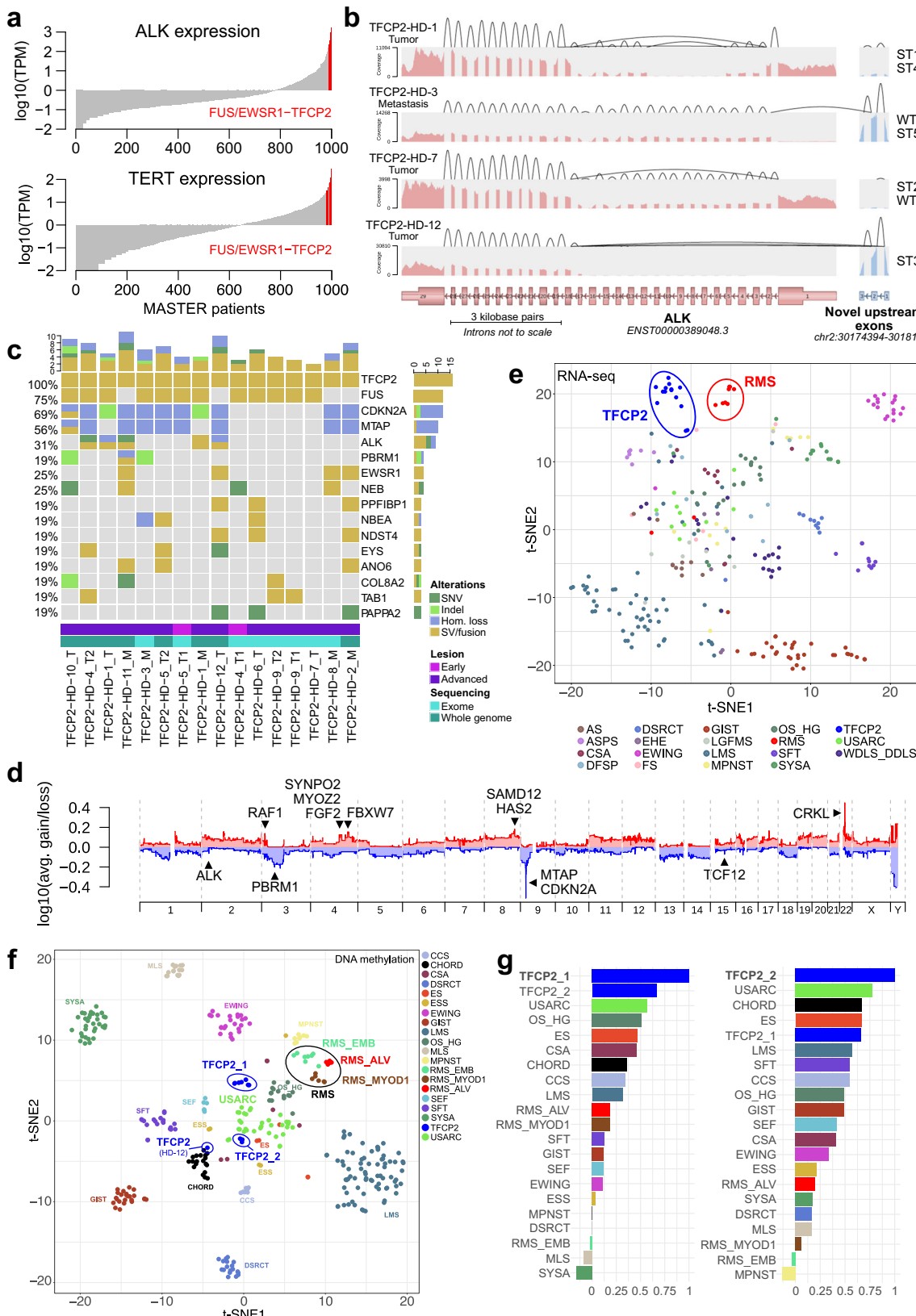

*CCND2* (Supplementary Data 1 and Supplementary Fig. 2f). Furthermore, we observed recurrent point mutations in *PAPPA2* (Fig. 2c and Supplementary Fig. 2g), which encodes the proteinase pappalysin 2 that cleaves insulin-like growth factor-binding protein 5 (IGFBP5)[32] known to play a role in myoblast proliferation, survival, and differentiation[33–35].

The tumors of the two patients with benign early lesions, which both carried a FUS-TFCP2 fusion, and malignant recurrences showed evidence of genomic evolution (Fig. 2c). In patient TFCP2-HD-4, the intragenic *ALK* deletion and homozygous *CDKN2A* loss were not detectable in the early tumor but in the local recurrence 28 months later. In patient TFCP2-HD-5, *CDKN2A* deletions were present at both

**Fig. 2 | Genetic characteristics of FUS/EWSR1-TFCP2 sarcoma. a** ALK and TERT mRNA expression of tumors from all patients enrolled in the MASTER program until November 21, 2018, and all FUS/EWSR1-TFCP2 cases from the MASTER and INFORM studies (indicated in red). TPM, transcripts per million. **b** Representative ALK transcript variants (see Supplementary Fig. 2c for the remaining cases). The expression levels of ALK exons are indicated by the depth of coverage (pink, regular exons; blue, upstream exons). The expression levels of splice junctions are indicated by the heights of the arcs connecting the exons. Intragenic deletions lead to exon skipping. **c** Genomic alterations occurring in three or more FUS/EWSR1-TFCP2 sarcoma samples. Copy-number aberrations were filtered for genes listed in the COSMIC Cancer Gene Census (https://cancer.sanger.ac.uk/census), and only homozygous deletions or amplifications with a total copy number above 2.5-fold base ploidy are shown. SNV, single-nucleotide variant; hom., homozygous; SV, structural variant. **d** Average copy-number profile across all TFCP2-rearranged cases. avg., average. **e** Dimensionality reduction (t-SNE) based on the expression levels of 792 transcription factors in 282 RNA-seq samples from 277 sarcoma patients, including 14 samples from 12 TFCP2-rearranged cases. **f** t-SNE using the 6,000 most variable CpG sites (mvCpGs) from 345 DNA methylation profiles of

sarcoma samples analyzed in MASTER ($n = 343$) and INFORM ($n = 2$). ESS combines samples assigned to methylation class ESS_HG or ESS_LG. CSA combines samples assigned to methylation class CSA_group_A, CSA_group_B, CSA_MES, or CSA_IDH_group_A. In addition, 11 FUS/EWSR1-TFCP2 cases from nine patients were included, which formed two clusters (TFCP2_1 and TFCP2_2), whereas one sample (HD-12) clustered separately. **g** Spearman correlation of TFCP2_1 (left panel) and TFCP2_2 (right panel) cases with samples from 19 sarcoma entities based on the same 6000 mvCpGs. Entities are sorted by decreasing correlation from top to bottom. The color code is the same as in **f**. TFCP2 FUS/EWSR1-TFCP2 sarcoma, RMS_ALV alveolar RMS, RMS_MYOD1 MYOD1-mutant spindle cell/sclerosing RMS, RMS_EMB embryonal RMS, MPNST malignant peripheral nerve sheath tumor, WDLS_DDLS well-differentiated and dedifferentiated liposarcoma, SFT solitary fibrous tumor, CCS clear cell sarcoma, ES epithelioid sarcoma, USARC undifferentiated sarcoma, CHORD chordoma, ASPS alveolar soft part sarcoma, SEF sclerosing epithelioid sarcoma, GIST gastrointestinal stromal tumor, AS angiosarcoma, CSA chondrosarcoma, OS_HG osteosarcoma high-grade, DSRCT desmoplastic small round cell tumor, ESS endometrial stromal sarcoma, EWING Ewing sarcoma, MLS myxoid liposarcoma, SYSA synovial sarcoma.

time points, 32 months apart, but additional mutations occurred in the advanced tumor. Moreover, the genomes of both cases developed a substantial degree of rearrangement during progression, with evidence of HRD in the advanced tumors (Supplementary Fig. 2e).

Visualization of RNA-seq data from the FUS/EWSR1-TFCP2 cases and 282 tumors comprising 19 different STS types using t-distributed stochastic neighbor embedding (t-SNE) showed clustering of the TFCP2-rearranged cases near but clearly separate from ARMS, ERMS, and PRMS cases (Fig. 2e). Among the top differentially expressed genes (DEG) between FUS/EWSR1-TFCP2 RMS and other RMS subtypes were ALK, TERT (encoding telomerase reverse transcriptase), and IGFBP5 and PAPPA2, further supporting the role of these two proteins in TFCP2-rearranged RMS (Supplementary Fig. 2h and Supplementary Data 2). TERT even showed outlier expression in FUS/EWSR1-TFCP2 tumors compared with all other MASTER cases (Fig. 2a). The DEG were associated with pathways or biological processes related to muscle (mainly downregulation of troponins and myosin heavy chains), keratinization (mainly upregulation of multiple keratins), and neuroactive ligand-receptor interactions (Supplementary Fig. 2i, j and Supplementary Data 2).

Finally, t-SNE analysis of 345 DNA methylation profiles of FUS/EWSR1-TFCP2 cases and other sarcomas comprising 23 methylation classes[36] revealed that the FUS/EWSR1-TFCP2 samples formed two distinct clusters (Fig. 2f). Unexpectedly, the TFCP2-rearranged cases were more closely related to undifferentiated sarcomas than other RMS subgroups. This observation was confirmed by global correlation analysis, suggesting that FUS/EWSR1-TFCP2 sarcoma is a distinct disease that may originate from a different cell of origin than other RMS subtypes (Fig. 2g and Supplementary Fig. 2k).

Taken together, TFCP2-rearranged sarcoma likely represents a separate entity characterized by distinct transcriptional and DNA methylation profiles, an unusual degree of genomic instability, truncated ALK variants, ALK and TERT overexpression, and recurrent *CDKN2A/MTAP* co-deletions.

### Structural and functional characteristics of FUS/EWSR1-TFCP2-associated ALK alterations

The frequent overexpression of wildtype (WT) ALK and/or ALK variants suggested that these events are critical for the pathogenesis of TFCP2-rearranged sarcoma. To systematically investigate this, we stably expressed the ALK variants in p53-deficient MCF10A human mammary epithelial cells, a widely used oncogenic transformation model (Supplementary Fig. 3a–d). The known oncogenic ALK alterations TPM3-ALK, EML4-ALK, ALK-F1147L, and ALK-ATI served as positive controls. ALK-ST1, ALK-ST2, and ALK-ST3 significantly increased EGF-independent colony formation of MCF10A cells, whereas ALK-

ST4, the only variant that lacks the kinase domain, and ALK-WT had no effect relative to empty vector (EV) (Fig. 3a and Supplementary Fig. 3e). ALK-ST1 and ALK-ST2 had stronger transforming capacity than ALK-ST3, although the latter was more highly expressed (Supplementary Fig. 3a). These results were substantiated by soft agar assays demonstrating the ability of ALK-ST1, ALK-ST2, and ALK-ST3 to confer anchorage-independent growth (Fig. 3b) and by xenotransplantations in immunocompromised mice showing rapid induction of tumor growth by the ALK variants with higher efficiency than ALK-ATI (Fig. 3c). Consistent with their oncogenic capacity, the three variants activated the MEK-ERK pathway. The strongest effect was exerted by ALK-ST2, which also induced AKT phosphorylation, whereas no variant activated STAT3 (Supplementary Fig. 3f). Likewise, immunohistochemistry of patient tumors showed consistent phosphorylation of ERK1/2 and PKCδ/θ and variable phosphorylation of STAT3 and AKT (Supplementary Fig. 3g and Supplementary Data 1). As expected, the three variants retaining the transmembrane domain were localized in the plasma membrane, whereas ALK-ATI showed cytoplasmic and nuclear staining as described[24] (Supplementary Fig. 3h). Since RMS is a mesenchymal malignancy and MCF10A cells are of epithelial origin, we validated these results in the immortalized mesenchymal stem cell line SCP-1, which may better reflect the epigenomic state of the yet unknown cell of origin of FUS/EWSR1-TFCP2 tumors (Supplementary Fig. 3i). ALK-ST1 and ALK-ST3, but not ALK-ST4, enhanced colony formation and induced anchorage-independent growth (Supplementary Fig. 3j, k). We failed to produce SCP-1 cells with stable overexpression of ALK-ST2 since they always died shortly after transduction, presumably due to oncogenic stress, as ALK-ST2 exhibited the strongest oncogenicity and pathway activation in MCF10A cells (Fig. 3a–c and Supplementary Fig. 3f).

As the oncogenic ALK-ST1, ALK-ST2, and ALK-ST3 proteins retained the catalytic domain, it was conceivable that their activity could be inhibited by clinical ALK inhibitors. Therefore, we tested the sensitivity of MCF10A cells expressing these ALK variants, ALK-WT, TPM3-ALK, or EV to the ALK inhibitors alectinib, ceritinib, and crizotinib in the absence of EGF. As expected, cells with ALK-WT or EV were not affected by ALK inhibition, whereas drug sensitivity was highest in the context of TPM3-ALK expression, followed by ALK-ST1, ALK-ST2, and ALK-ST3 (Fig. 3d). Furthermore, the three inhibitors showed varying efficacy against the oncogenic ALK variants, with ceritinib being the most effective, followed by crizotinib and alectinib (Fig. 3d). We also tested the sensitivity of viable tumor cells carrying ALK-ST2 and ALK-ST3 from patient TFCP2-HD-4 to these ALK inhibitors in short-term cultures. While the cells showed moderate or no response to crizotinib (half-maximal inhibitory concentration [$IC_{50}$], 1.62 μM) and alectinib ($IC_{50}$, 26.64 μM), respectively, they were highly sensitive to

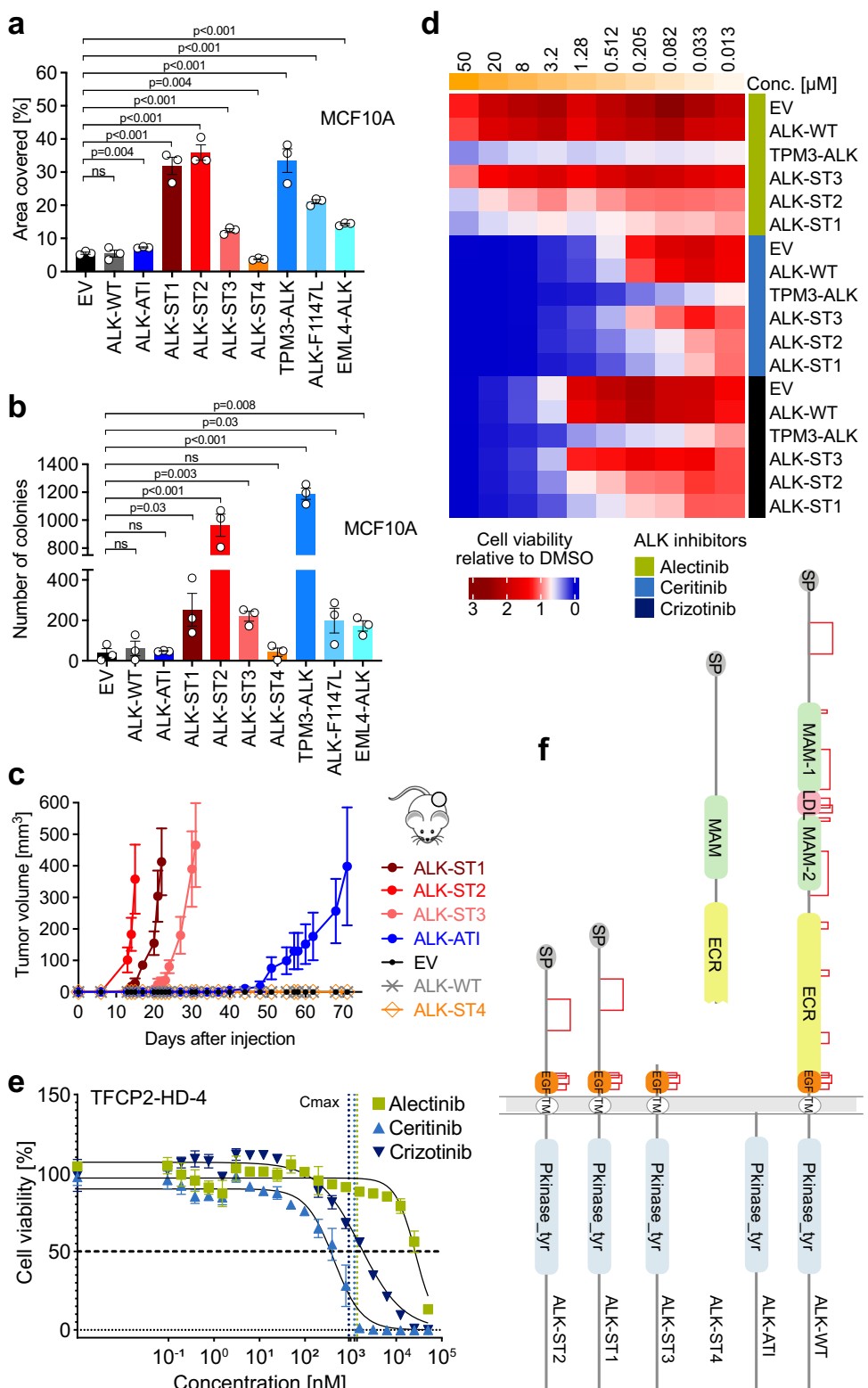

ceritinib (IC$_{50}$, 0.43 μM) (Fig. 3e), consistent with the results in MCF10A cells. These data indicated that the FUS/EWSR1-TFCP2-associated ALK alterations are, in principle, targetable, albeit with varying sensitivity to different drugs.

The functional properties of the ALK variants suggested varying transformation potencies (ALK-ST2 > ALK-ST1 > ALK-ST3 > ALK-ATI), which was intriguing as the truncated ALK proteins lack all known functional domains of the extracellular moiety while retaining supposedly non-functional extracellular segments and the

transmembrane and intracellular domains (Supplementary Fig. 2d). Particularly ALK-ST1 and ALK-ST2 differ only in 40 aa outside any known functional domain. This raised the question of whether previously uncharacterized structures are involved in the regulation of ALK activity. To address this, we performed structural modeling of ALK-WT and ALK variants using HHpred[37], which allows the prediction of structural regions and intra- and intermolecular interactions (Fig. 3f and Supplementary Fig. 3l). Human ALK contains a signal peptide, two extracellular MAM domains interspersed with an LDL receptor class A

**Fig. 3 | Functional and structural characteristics of ALK alterations associated with FUS/EWSR1-TFCP2 RMS. a** Colony formation of MCF10A cells stably transduced with ALK alterations or EV. Mean ± SEM (*n* = 3 independent experiments). **b** Anchorage-independent growth in soft agar of MCF10A cells stably transduced with ALK alterations or EV. Mean ± SEM (*n* = 3 independent experiments). **c** Tumor growth in NOD-SCID mice of MCF10A cells stably transduced with ALK alterations or EV. Shown is the mean volume ± SEM of six tumors per cell line until the first mouse had to be sacrificed in one group due to reaching the maximum allowed tumor length. **d** Sensitivity of MCF10A cells stably transduced with the indicated ALK variants or EV to crizotinib, ceritinib, or alectinib after 72 h in the absence of EGF. Drug concentrations (conc.) are shown at the top of the heatmap. Mean cell viability of drug-treated cells relative to the respective DMSO control (*n* = 2 independent experiments). **e** Sensitivity of freshly isolated and singularized cells from tumor sample TFCP2-HD-4 to the ALK inhibitors crizotinib, ceritinib, or alectinib. Vertical dotted lines represent the $C_{max}$ of each compound. Mean ± SEM (*n* = 4 technical replicates). **f** Domain architectures of ALK variants. Pairings between cysteines in each variant are indicated by red lines. SP signal peptide, MAM meprin A-5 protein, and receptor protein-tyrosine phosphatase mu domain, LDL low-density lipoprotein receptor class A, EGF epidermal growth factor-like domain, TM transmembrane helix, Pkinase_tyr tyrosine protein kinase domain. Statistical significance was assessed by a one-tailed unpaired *t*-test. ns not significant. Source data for **a**–**e** are provided in the Source Data file.

followed by a ligand-binding extracellular region (ECR) consisting of a glycine-rich region and an EGF-like domain, a transmembrane helix region, and an intracellular tyrosine kinase domain[38]. We identified 26 cysteines in the extracellular segment, two of which are outside the well-studied domains and present in ALK-ST1, ALK-ST2, and ALK-WT but not ALK-ST3 (Fig. 3f and Supplementary Fig. 3l). All cysteines were paired according to the DIpro[39] disulfide bridge prediction server (Fig. 3f), and it seemed possible that some of them form intermolecular disulfide bonds that lead to the hyperactivation of ALK variants. This notion was supported by a previous study that used an IgG2b Fc fragment-ALK fusion protein to drive kinase activation through the formation of intermolecular disulfide bonds[40]. Moreover, the EGF-like domain of the ECR, in combination with the kinase domain, might enhance the activity of ALK-ST1, ALK-ST2, and ALK-ST3 compared to ALK-ATI, while the additional signal peptide and extracellular segments might further increase the activity of ALK-ST1 and ALK-ST2 relative to ALK-ST3.

Together, these data demonstrated that ALK-ST1, ALK-ST2, and ALK-ST3 have similar or stronger transforming potential than known oncogenic ALK alterations and are, in principle, targetable by kinase inhibitors. An EGF-like domain and disulfide bonds in the extracellular part of the transforming ALK variants may influence receptor activity as predicted by structural modeling.

### TFCP2 fusions block late myogenic differentiation

We next investigated the oncogenicity of the FUS/EWSR1-TFCP2 fusions. Stable expression of FUS-TFCP2 did not enhance colony formation on plastic of MCF10A and SCP-1 cells (Supplementary Fig. 4a–d). Also, immunocompromised mice subcutaneously injected with FUS-TFCP2-transduced MCF10A cells did not develop tumors after an observation period of 300 days.

To test whether FUS-TFCP2 might affect myogenic differentiation, we employed the immortalized human skeletal muscle cell line LHCN-M2 derived from myoblasts by ectopic expression of hTERT and CDK4[41]. These cells grow as undifferentiated myoblasts in high-serum conditions but rapidly differentiate and fuse into contractile myotubes in low serum (Fig. 4a). Stable expression of FUS-TFCP2 delayed differentiation of LHCN-M2 cells under low-serum conditions, as evidenced by fewer nuclei per myotube, i.e., a smaller fusion index, compared to cells transduced with EV, TFCP2, or FUS (Fig. 4b, c and Supplementary Fig. 4e–g). Consistent with this observation, the expression pattern of the transcription factors MYOD and MYOG, which are essential for myogenic differentiation, was altered. Instead of the normally occurring early-phase increase and subsequent slow decrease of MYOD, FUS-TFCP2 resulted in a delayed increase in MYOD that then persisted (Fig. 4a, d). The late phase of differentiation is initiated by a MYOG increase as an essential step for myocyte fusion, which was evident in EV cells but delayed and insufficient in FUS-TFCP2-expressing cells (Fig. 4a, d). Thus, the impaired differentiation of LHCN-M2 cells expressing FUS-TFCP2, which did not form myotubes even after a prolonged time in the differentiation medium (Fig. 4e), was likely due to their inability to downregulate MYOD and insufficient MYOG expression.

Collectively, these observations showed that FUS-TFCP2, although not transforming on its own, can confer the ability of myoblasts to block late myogenic differentiation.

### FUS/EWSR1-TFCP2 cause widespread transcriptional changes including upregulation of ALK

Based on the association of FUS/EWSR1-TFCP2 and ALK overexpression, we speculated that ALK might be transcriptionally activated by the fusions. Indeed, FUS-TFCP2 and, to a lesser extent, EWSR1-TFCP2 increased ALK mRNA and protein in MCF10A, SCP-1, and LHCN-M2 cells (Fig. 5a; Supplementary Fig. 4a, b and Supplementary Fig. 5a, b). To find additional FUS/EWSR1-TFCP2 target genes, we performed RNA-seq with MCF10A cells stably expressing the fusions, full-length fusion partners, or EV. Both fusions and the three WT genes altered the expression of several genes relative to EV (Supplementary Fig. 5c, d and Supplementary Data 3). FUS-TFCP2 and EWSR1-TFCP2 mainly activated genes (81% and 84% of deregulated genes, respectively), whereas FUS, EWSR1, and TFCP2 predominantly repressed transcription (83%, 68%, and 73% of deregulated genes, respectively) (Supplementary Fig. 5d). *CCND2* was the only gene significantly repressed by both fusions and TFCP2 (Supplementary Fig. 5c, d). These observations indicated that FUS-TFCP2 and EWSR1-TFCP2 acquired activating transcriptional properties not inherent to TFCP2.

To narrow down target genes of FUS/EWSR1-TFCP2 critical for malignant transformation, we also performed RNA-seq with SCP-1 cells transduced with either fusion, the WT fusion partners, or EV. Both fusions, especially EWSR1-TFCP2, had a substantial impact on the transcriptional landscape of SCP-1 cells, with 931 (FUS-TFCP2) and 3,522 (EWSR1-TFCP2) genes significantly deregulated (Supplementary Data 3). The overlap of genes deregulated by both fusions in SCP-1 and MCF10A cells was modest, with only CCND2 significantly differentially expressed in all four conditions and 16 genes deregulated in three of the four samples (Fig. 5b, c and Supplementary Fig. 5e). The expression pattern of these 17 genes was largely reflected in the patient samples (Fig. 5d). Three genes, i.e., *HSD11B2*, *IGFBP5*, and *PTH1R*, were more highly expressed in TFCP2-rearranged patient samples than in 90% of tumors analyzed in the MASTER program (Fig. 5d). *HSD11B2* encodes hydroxysteroid 11-beta dehydrogenase 2, which catalyzes the conversion of cortisol to cortisone. *PTH1R* encodes the parathyroid 1 receptor and, like IGFBP5, was also among the DEG upregulated in FUS/EWSR1-TFCP2 sarcoma compared with other RMS types (Supplementary Fig. 2h). Both IGFBP5 and PTHR1 are involved in muscle cell proliferation and differentiation[42,43].

Together, these data showed that FUS/EWSR1-TFCP2 transcriptionally activates a core set of genes that include ALK and regulators of muscle physiology, which together may play a role in sarcoma development.

### FUS-TFCP2 binds to *ALK* and *TERT* gene loci

To determine whether the transcriptional effects of FUS-TFCP2 were mediated by direct binding to DNA, we performed antibody-guided chromatin tagmentation sequencing (ACT-seq) with two different anti-HA antibodies and an anti-H3K27ac antibody in MCF10A cells

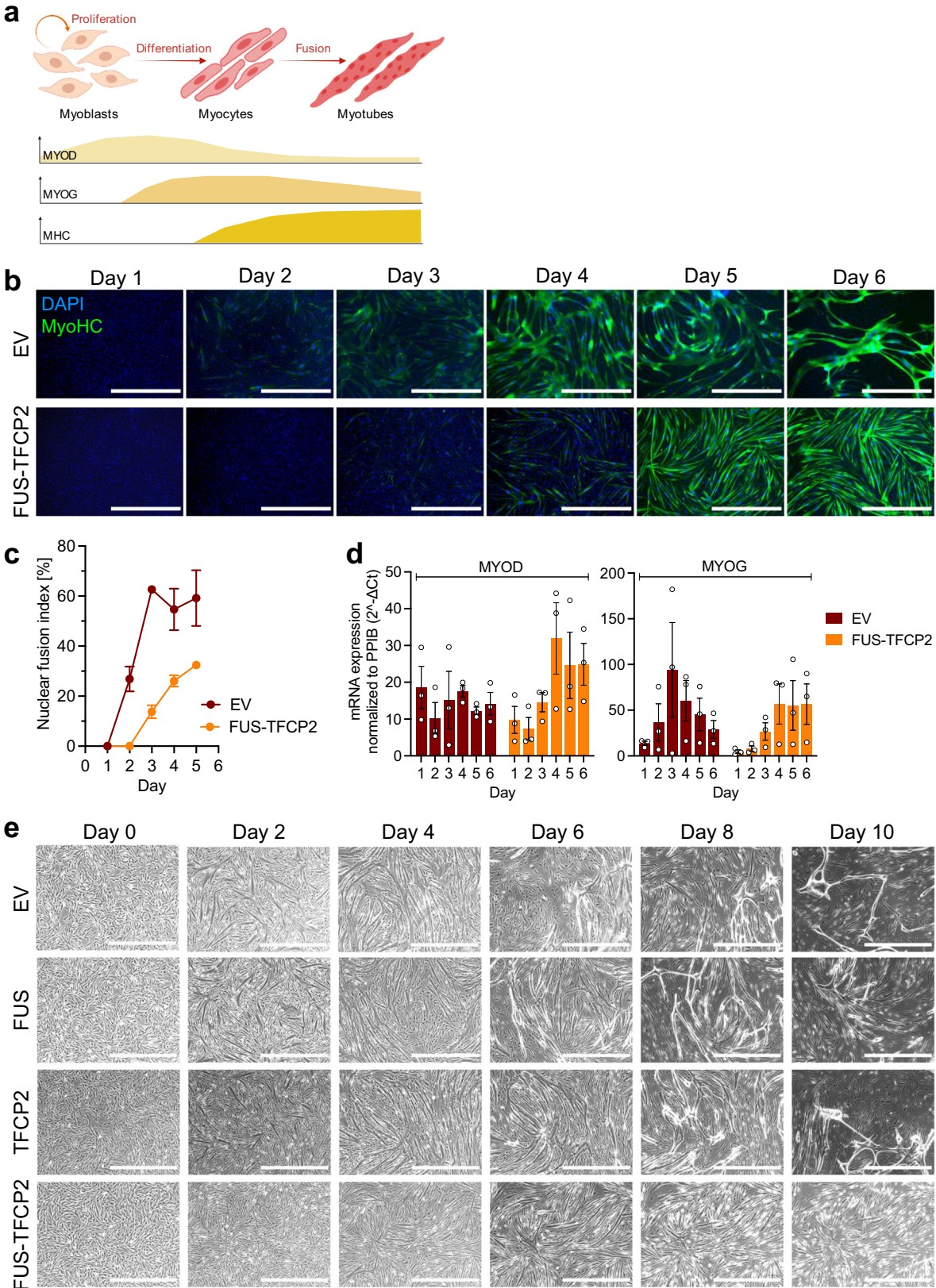

stably expressing EV, HA-tagged TFCP2 (HA-TFCP2), or HA-FUS-TFCP2. Despite the low complexity of the libraries, which can lead to false negative sites in the genome, we were able to predict the transcription factor-binding motif of TFCP2 from the ACT-seq data, which consisted of 15 nucleotides and was nearly identical to that of the mouse ortholog Tcfcp2l1 inferred from chromatin immunoprecipitation sequencing data in the HOMER database[44] (Supplementary Fig. 5f). The predicted binding motif of FUS-TFCP2 contained the same sequence but with an additional repeat of nine nucleotides at the 3' end (Supplementary Fig. 5f). This suggested that the detected peaks are reliable, although their numbers might be underestimated.

**Fig. 4 | Effects of TFCP2 fusions on myogenic differentiation. a** Stages of muscle cell differentiation from myoblasts to myotubes and corresponding expression pattern of MYOD and MYOG transcription factors and myosin heavy chain (MHC). The image was created with BioRender.com and adapted from Bentzinger et al.[64]. **b** Immunofluorescence images of LHCN-M2 cells transduced with EV or FUS-TFCP2 and cultured in differentiation medium for six days. Green, MyoHC; blue, DAPI (nuclei). Representative images of three independent experiments with similar results are shown. Scale bar, 1 mm. **c** Fusion index of LHCN-M2 cells transduced with EV or FUS-TFCP2 and cultured in differentiation medium for five days. Mean ± SEM (*n* = 3 independent experiments). **d** Relative MYOD and MYOG mRNA expression of LHCN-M2 cells transduced with EV or FUS-TFCP2 and cultured in differentiation medium for six days. Mean ± SEM (*n* = 3 independent experiments). **e** Phase-contrast images of LHCN-M2 cells transduced with EV, FUS, TFCP2, or FUS-TFCP2 and cultured in differentiation medium for ten days. Representative images of six independent experiments with similar results are shown. Scale bar, 1 mm. Source data for **c** and **d** are provided in the Source Data file.

Inspection of specific gene loci revealed that TFCP2 and FUS-TFCP2 bound to the TFCP2 binding motif CAGCCCTGTCCAGT CCAGTT in a genomic region 45 kilobases (kb) upstream of the *ALK* transcriptional start site and 9 kb upstream of the unannotated *ALK* exons −3, −2, and −1, where the transcriptional activation mark H3K27ac appeared to be enriched (Fig. 5e and Supplementary Fig. 5g). This suggested that FUS-TFCP2 can directly regulate *ALK* transcription, possibly through an alternative promoter upstream of the unannotated *ALK* exons, explaining their expression in most TFCP2-rearranged tumors. Consistent with the observation that TFCP2 bound to the same region, we found expression of ALK exons −3, −2, and −1 in 3% of non-FUS/EWSR1-TFCP2 cases analyzed in the MASTER program, which correlated highly with TFCP2 mRNA levels, even more strongly than with ALK itself, suggesting a regulatory role of TFCP2 in the transcription of the unannotated exons (Supplementary Fig. 5h, i).

Finally, we observed prominent binding of FUS-TFCP2 and TFCP2 in a cluster of TFCP2 binding motifs in the intron before the third coding exon of *TERT* (Fig. 5f and Supplementary Fig. 5j), explaining the extreme TERT expression in TFCP2-rearranged patient samples (Fig. 2a). Specifically, TFCP2-rearranged sarcoma expressed a TERT variant lacking exons 1 and 2 (Δex1-2), and intronic RNAs were transcribed in the reverse direction, starting exactly at the TFCP2 binding region (Fig. 5g). The shorter transcript, consisting of TERT exons 3 to 16, contains an open reading frame encoding a 584-aa TERT protein that harbors the reverse transcriptase catalytic domain but lacks the main part of the RNA-binding domain (Fig. 5h). Accordingly, we detected high expression of a short TERT protein in TFCP2-rearranged tumors, which was absent in HeLa control cells (Fig. 5i). To gain further evidence for transcriptional regulation of *TERT* by TFCP2 fusions, we went back to the RNA-seq data from the isogenic cell lines and found significantly induced TERT expression in SCP-1 cells transduced with FUS-TFCP2, EWSR1-TFCP2, or TFCP2, but not in MCF10A cells, suggesting a cell lineage-dependent influence (Supplementary Data 3).

Together, these data suggested that FUS-TFCP2 contributes to malignant transformation by transcriptionally activating the *ALK* oncogene and a TERT Δex1-2 variant through binding to genomic regions outside their regular promoters.

### FUS/EWSR1-TFCP2 affect DNA repair

In contrast to other fusion-driven sarcomas, FUS/EWSR1-TFCP2-positive tumors exhibited substantially rearranged genomes, suggesting a defect in DNA repair. In support of this, a reported HRD-associated gene signature[45] was enriched in MCF10A and SCP-1 cells expressing FUS-TFCP2 or EWSR1-TFCP2 compared with EV, TFCP2, and FUS (Fig. 6a). MCF10A cells treated with the DNA-damaging agent cisplatin showed significantly lower viability when expressing FUS-TFCP2 than cells transduced with EV, FUS, or TFCP2, which was accompanied by the induction of caspase 3/7 activity, indicating enhanced apoptosis (Fig. 6b, c). We next quantified γH2AX formation as a direct indicator of DNA double-strand breaks in transduced MCF10A cells. The presence of FUS-TFCP2 alone increased the number of cells with high γH2AX levels as determined by flow cytometry (Fig. 6d, left panel; Supplementary Fig. 6a) and the number of γH2AX foci per cell as determined by immunofluorescence (Fig. 6e, left panel). Treatment with cisplatin for 4 h elevated γH2AX levels and foci numbers in MCF10A cells transduced with EV, FUS, TFCP2, or FUS-TFCP2 to a

similar extent, such that higher levels were maintained in the presence of the fusion (Fig. 6d, e, middle panel). To gain insight into the DNA repair capacity of the cells, we removed cisplatin, followed by culture for an additional 24 h in a standard medium. Whereas the cells transduced with EV or the WT fusion partners showed either declining or unchanged DNA double-strand breaks, the FUS-TFCP2-positive cells had further increasing γH2AX levels, indicating impaired DNA repair (Fig. 6d, e, right panel; Fig. 6f; Supplementary Fig. 6b).

Together, these data suggested that TFCP2 fusions contribute to the development of genomically unstable sarcoma by impairing the DNA repair machinery.

## Discussion

We here describe the clinical and histopathologic characteristics of sarcomas with FUS/EWSR1-TFCP2 fusions, which were categorized as spindle cell/sclerosing RMS in the most recent WHO classification, provide a comprehensive multi-omics portrait of these tumors, including two benign precursor lesions, and present the oncogenic properties and associated therapeutic vulnerabilities of underlying genetic alterations.

The tumors affected children and adults and occurred primarily in the head and neck region, with a predilection for craniofacial bones, consistent with previous studies[9]. Of the cases reported to date, few patients were alive without evidence of disease after 20 to 21 (*n* = 3)[9] or 108 months (*n* = 1)[46]. In our cohort, all patients relapsed or progressed after a median of 6.5 months, with the latest relapse occurring 28 months after primary treatment. These numbers and the striking failure of numerous chemotherapy regimens indicate that TCFP2-rearranged sarcomas are uniformly fatal and not adequately treated with standard sarcoma protocols. Despite the rapid relapses and lack of effective systemic therapies, the median overall survival was 29 months, which may be explained by repeated surgical procedures in patients with locoregional relapse.

The tumors' morphologic and immunohistochemical features were consistent with previously reported cases[6–12,46–48]. Most were mixed spindle cell/epithelioid neoplasms and expressed cytokeratin (AE1/E3) in addition to the RMS-defining factors MYOD1 and/or MYOG. Furthermore, many tumors showed strong ALK expression. An important aspect beyond what has been described previously is the rather bland morphologic pattern of early lesions that were considered benign, showing spindle cell infiltration of the dermis and subcutaneous fat with only slightly increased cellularity and minimal to low nuclear atypia, making the diagnosis of sarcoma difficult.

Unlike other gene fusion-related sarcomas, the FUS/EWSR1-TFCP2 cases showed highly rearranged genomes and thus were more reminiscent of, e.g., undifferentiated sarcoma. DNA methylation analysis supported this relationship, indicating that the categorization of sarcomas with TFCP2 fusions as RMS should be revisited and underscoring the potential of multilayered molecular profiling to refine and sometimes revise conventionally defined disease categories. In addition to genomic instability and entity-defining TFCP2 fusions, we identified several distinct genetic and transcriptional characteristics. The most prominent alterations involved the ALK RTK, which is affected by oncogenic translocations or point mutations in various cancers. In nearly all FUS/EWSR1-TFCP2 tumors, ALK is highly expressed, as previously noted[6–12]. This applies not only to ALK-WT but also to

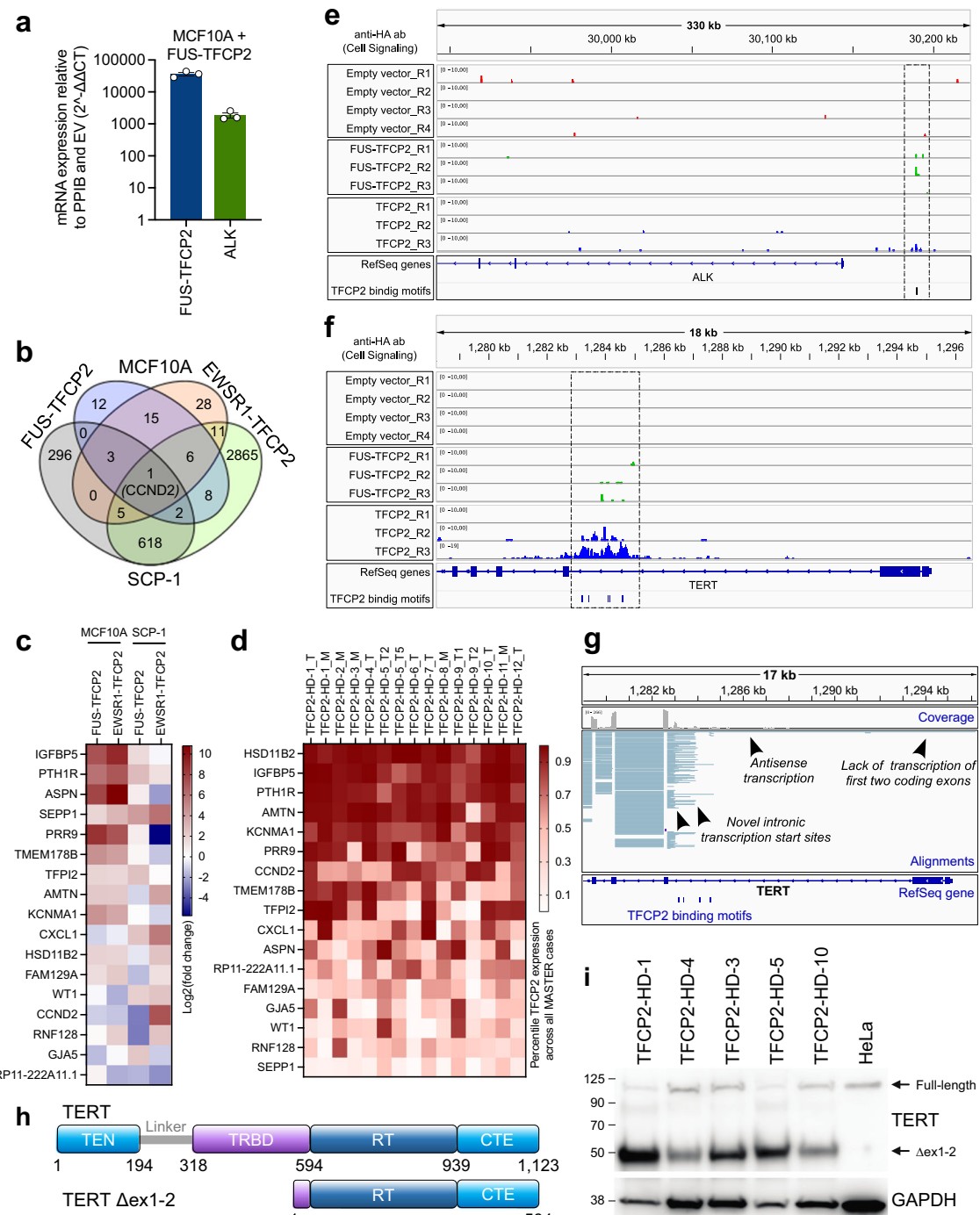

**Fig. 5 | Transcriptional effects of TFCP2 fusions. a** Relative FUS-TFCP2 and ALK mRNA expression in MCF10A cells stably transduced with FUS-TFCP2. Mean ± SEM (*n* = 3 independently transduced cell lines). Source data are provided in the Source Data file. **b** Number of genes significantly deregulated in MCF10A and SCP-1 cells transduced with FUS-TFCP2 or EWSR1-TFCP2 versus cells transduced with EV (log2(fold-change) >1.0 or <−1.0), as determined by RNA-seq. **c** Genes significantly deregulated in MCF10A and SCP-1 cells transduced with FUS-TFCP2 or EWSR1-TFCP2 versus cells transduced with EV (log2(fold-change) >1.0 or <−1.0 in at least three cell lines), as determined by RNA-seq. **d** Expression of genes from **c** in FUS/ EWSR1-TFCP2-positive sarcoma samples, indicated as percentiles of expression across the entire MASTER cohort. **e, f** Genome browser images of *ALK* (**e**) and *TERT* (**f**) showing enrichment peaks obtained by ACT-seq with an anti-HA antibody (Cell Signaling) in MCF10A cells stably expressing EV, HA-TFCP2, or HA-FUS-TFCP2.

**g** Genome browser image showing aberrant transcription originating from the second intron of *TERT* in patient TFCP2-HD-1. The first two exons were not transcribed. Instead, multiple novel intronic transcription start sites (red alignments) and antisense transcription (blue reads) were found around putative TFCP2 binding sites (bottom) detected by HOMER using the known Tcfcp2l1 binding motif.
**h** Domain structure of full-length TERT (top) and the TERT variant predicted to be translated from mRNA lacking exons 1 and 2 (bottom). The domain structure was adapted from Chan et al.[65]. CTE C-terminal extension, RT reverse transcriptase, TEN TERT-essential N-terminal, TRBD telomerase RNA-binding domain. **i** Western blot with tumor tissue from five patients and HeLa cells with an antibody binding to the C-terminus of TERT. Protein masses in kDa are shown on the left. Uncropped blots are provided in the Source Data file.

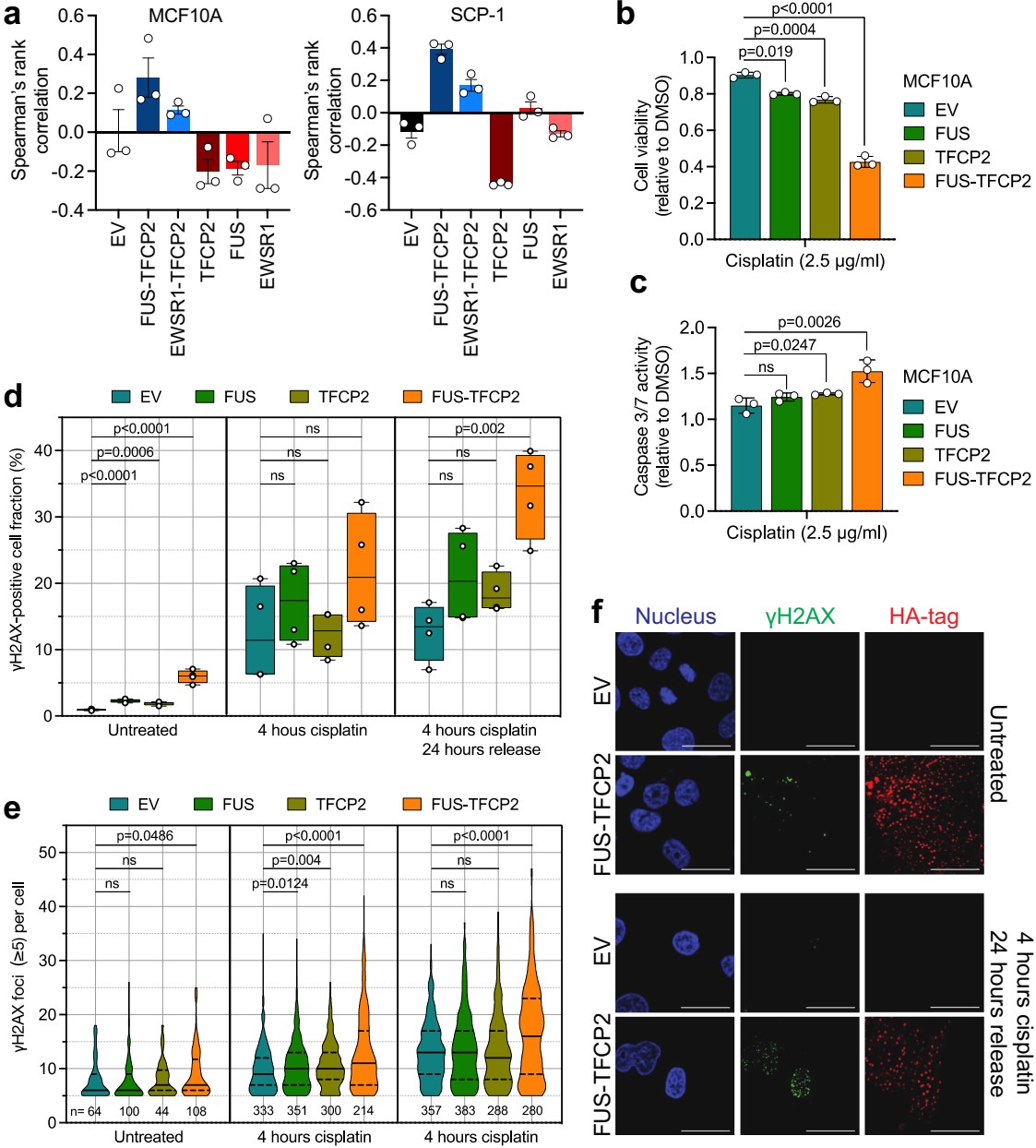

**Fig. 6 | Impaired DNA repair in FUS/EWRS1-TFCP2-expressing cells.**
**a** Correlation of the gene expression profiles induced by EV, FUS-TFCP2, EWSR1-TFCP2, TFCP2, FUS, or EWSR1 in MCF10A (left panel) and SCP-1 (right panel) cells with the HRD gene expression signature reported by Peng et al.[45]. A positive Spearman rank correlation value indicates HRD. Mean ± SEM (n = 3 independently transduced cell lines). **b**, **c** Effect of cisplatin on cell viability after six days of treatment (b) and caspase 3/7 activity after three days of treatment (c) in MCF10A cells expressing EV, FUS, TFCP2, or FUS-TFCP2. Mean ± SD (n = 3 independent experiments). **d** Quantification of γH2AX levels by flow cytometry in MCF10A cells expressing EV, FUS, TFCP2, or FUS-TFCP2. Cells were cultured in regular medium (left panel), incubated with 2.5 μg/ml cisplatin for 4 h (middle panel), or treated with 2.5 μg/ml cisplatin for 4 h followed by a 24-h incubation in regular medium (right panel). Box and whisker plot (median; box, 25th to 75th percentiles; whiskers, min to max; n = 4 independent experiments). **e** Violin plots (lines: median and quartiles) showing the number of γH2AX foci per cell determined by immunofluorescence. The same cells as in **d** were used from n = 2 independent experiments, and only cells with five or more foci per cell are shown. The number of cells per condition is indicated at the bottom. **f** Representative immunofluorescence images of MCF10A cells from the experiment shown in **e**. Additional images are provided in Supplementary Fig. 6. Scale bar, 100 μm. Statistical significance was assessed by a two-tailed unpaired t-test. ns, not significant. Source data for **a**–**e** are provided in the Source Data file.

short variants caused by intragenic deletions, alternative transcription, and aberrant splicing. Intragenic deletions have been previously observed but not further characterized, and it has been speculated that they might cause high ALK expression[9,11,46]. Our data rather suggest that ALK transcription is driven by the direct binding of FUS-TFCP2 to *ALK* upstream of its promoter, a conclusion further supported by the identification of ALK transcripts in which exon 1 is replaced by two exons upstream of the regular transcriptional start

site. The high ALK expression may contribute to the occurrence of oncogenic ALK deletions through transcription-associated mutagenesis[49].

Three of the discovered truncated ALK receptors, which retain the transmembrane and kinase domains, transformed immortalized cells with comparable or greater potency than known oncogenic ALK variants. Thus, ALK likely contributes to the development of FUS/EWSR1-TFCP2 sarcomas and could, in principle, be a therapeutic target. This

was supported by ex vivo experiments with patient-derived tumor cells carrying the ALK-ST2 and -ST3 variants and with isogenic MCF10A cells, both of which were sensitive to ceritinib and crizotinib but not or to a much lesser extent to alectinib. Also, the different variants conferred varying sensitivity to ALK inhibition, which was high for ALK-ST1 and ALK-ST2 but low, comparable to that of ALK-WT, for ALK-ST3. However, it must be taken into account that this analysis was performed in the absence of ALK ligand, which may be different in patient tumors, so ALK-WT and ALK-ST3 may also be responsive targets in vivo.

To date, two patients with FUS/EWSR1-TFCP2 sarcoma who received ALK inhibitors have been reported. One received crizotinib for four weeks and showed stable disease of the primary tumor but progressive lung metastases[50]. The tumor cells in this case carried a deletion of ALK exons 2–16 resulting in the expression of ALK-ST2. The other underwent radiotherapy, leading to stable disease, followed by treatment with crizotinib for one month and subsequent alectinib, which resulted in tumor shrinkage. Progression 11 months later was halted by lorlatinib[51]. This patient's tumor cells showed high ALK expression, but it remained unclear if an ALK variant was present. In our study, six patients received ALK inhibitors with palliative intent, and response data were available in three. Of those, two patients with tumors expressing ALK-ST1 or ALK-WT were treated with crizotinib, resulting in disease progression and a mixed response, respectively. The tumor in the third patient expressed ALK-WT and progressed after therapy with alectinib, prompting a switch to lorlatininb. However, this patient and the remaining three treated with crizotinib or ceritinib experienced rapid clinical deterioration and died before imaging studies could be performed after an average of 1.5 months. All patients, including previously reported cases, were resistant to multiple lines of drug treatment. It is likely that extensive prior therapies accelerate tumor evolution and promote the selection of particularly aggressive cell clones, which could diminish response to ALK inhibition. In line with this, we observed that cultured tumor cells from patient TFCP2-HD-4 obtained after surgery and adjuvant radiation therapy but before systemic therapy were highly sensitive to ceritinib. However, the patient was treated with ceritinib at a very late disease stage after receiving chemotherapy, the multi-kinase inhibitor pazopanib, and repeat radiotherapy, and response assessment was no longer feasible due to the patient's rapid deterioration and death. Despite the limited number of cases, the observations in patients and ex vivo and in vitro drug response data support that ALK inhibitors should be further explored in FUS/EWSR1-TFCP2 sarcoma. However, they should be tested in therapy-naïve patients and/or as part of rational combination regimens to fully evaluate their clinical potential. Furthermore, both the underlying ALK variant and the ALK inhibitor administered appear relevant.

The second most common alteration in FUS/EWSR1-TFCP2 sarcomas, with an incidence of 75%, were deletions or indels of the *CDKN2A* tumor suppressor, which have been noted previously[8,9,11,46]. *CDKN2A* was always co-deleted with the neighboring *MTAP* gene, which encodes 5-methylthioadenosine phosphorylase, a critical enzyme in the adenine and methionine salvage pathways. Two studies have shown that MTAP loss leads to accumulation of the metabolite methylthioadenosine, which inhibits the catalytic function of protein methyltransferase 5 (PRMT5), resulting in sensitivity to pharmacologic inhibition of residual PRMT5 activity[30,31]. The efficacy of PRMT5 inhibitors in the context of *MTAP* passenger deletions has been demonstrated in several cancers[30,31], suggesting that patients with FUS/EWSR1-TFCP2 sarcoma and CDKN2A/MTAP loss may also benefit from PRMT5 inhibitors, which are currently being tested in clinical trials.

The fourth gene deregulated in TFCP2-rearranged sarcoma, besides *ALK*, *CDKN2A*, and *MTAP*, is *TERT*, encoding the catalytic component of the telomerase complex that protects dividing cells from progressive telomere shortening, subsequent senescence, and

malignant transformation[52]. TERT is aberrantly expressed in up to 90% of cancers by various mechanisms, including promoter mutations, amplification, and binding of activating transcription factors to its promoter[52]. We found very high expression of a shortened TERT mRNA lacking exons 1 and 2 in all TFCP2-rearranged cases, caused by the binding of FUS-TFCP2 to intronic regions immediately upstream of exon 3, translating into the expression of a smaller TERT protein. This truncated TERT protein contains the reverse transcriptase catalytic domain but lacks the main part of the high-affinity telomerase RNA-binding domain and the N-terminal TEN domain, which are both required for full telomerase activity[53]. It is, therefore, questionable whether it can fulfill its role as a telomere-extending enzyme. However, TERT was shown to have non-canonical functions as well[54]. For example, full-length TERT and the Δ4-13 splice variant promote cell proliferation[54,55], and TERT has been implicated in DNA repair (see below). The exact function of TERT Δex1-2 remains to be determined.

The TFCP2 fusions blocked late myogenic differentiation, a property not inherent to TFCP2. Several lines of evidence indicated the involvement of IGFBP5, a multifunctional protein that acts dependent or independent of insulin growth factor (IGF) signaling and plays an essential role in myoblast proliferation, survival, and differentiation[42]. First, we found high expression of IGFBP5 in TFCP2-rearranged sarcoma. Second, IGFBP5 was the most upregulated gene in MCF10A and SCP-1 cells expressing either TFCP2 fusion but was not induced by TFCP2. Third, *PAPPA2*, encoding the IGFBP5-specific proteinase pappalysin 2, was among the recurrently mutated genes in FUS/EWSR1-TFCP2 sarcoma. Furthermore, *PTH1R*, encoding the parathyroid 1 receptor, was also highly expressed in FUS/EWSR1-TFCP2 sarcomas and strongly induced by TFCP2 fusions in immortalized cells. A link between PTH1R and IGFBP5 was demonstrated by the finding that stimulation of osteosarcoma cells with parathyroid hormone, the ligand of PTHR1, induced IGFBP5 expression[56], suggesting another mechanism for maintaining high IGFBP5 levels. If the PAPPA2 mutations were inactivating, IGFBP5 protein levels would increase through impaired degradation, consistent with observations in Pappa2 knockout mice[32,57]. However, they could also increase catalytic activity and lead to the augmented generation of N- and C-terminal IGFBP5 fragments, which have various effects on signaling pathways and cellular properties, e.g., IGF-independent enhancement of vascular smooth muscle cell migration[42]. Whether the role of IGFBP5 in TFCP2-rearranged sarcoma depends on IGF signaling and is mediated via the full-length protein or proteolytic fragments remains to be determined.

Finally, we found evidence of deficient DNA repair in the context of FUS/EWSR1-TFCP2 fusions, as indicated by the highly rearranged genomes of patient tumors, the enrichment of an HRD-associated gene signature in immortalized cell lines expressing FUS-TFCP2 or EWSR1-TFCP2, and the reduced viability and delayed DNA double-strand break repair of FUS-TFCP2-positive cells following cisplatin treatment. One contributing factor may be the lack of FUS, which localizes to sites of DNA damage and is required for efficient repair of DNA double-strand breaks by HR or non-homologous end joining[15,18], functions that require C-terminal domains lost by fusion with TFCP2. In addition, the very high expression of the TERT Δex1-2 variant, which most likely lacks telomere-elongating activity and therefore may be equivalent to TERT loss, could contribute to HRD. For example, cells with persistent TERT loss were shown to exhibit altered overall chromatin structure, impaired activation of the DNA damage response, enhanced radio-sensitivity, and increased numbers of chromosome fragments upon irradiation[54,58]. The clinical course of the patient described above who responded to radiotherapy followed by ALK inhibition also suggests that these tumors have reduced DNA repair capacity[51]. Zein et al. reported a patient with RMS carrying an EWSR1-UBP1 fusion, which seems to be a variant of EWSR1-TFCP2, as TFCP2 and UBP1 belong to the same transcription factor family, have high sequence homology,

and share the same domain structure[59]. This patient responded to a lung cancer regimen that included carboplatin, which is not a standard in RMS, but progressed on subsequent maintenance therapy with pembrolizumab[59].

Collectively, our data show that TFCP2-rearranged neoplasms represent an unusual disease whose molecular landscape suggests categorization as undifferentiated sarcoma rather than RMS. They perhaps arise from a different cell of origin than RMS and seem to develop gradually from precursor lesions, a feature known primarily in carcinomas. Furthermore, we shed light on the remarkably versatile role of TFCP2 fusions as truncal driver alterations that interfere with myogenic differentiation and promote various secondary molecular changes, such as induction of ALK and overexpression of a TERT Δex1-2 variant, which might have non-canonical oncogenic functions. Finally, our findings suggest that patients with TFCP2-rearranged sarcomas, which are refractory to conventional chemotherapy, may benefit from combination treatments that include platinum derivatives, ALK inhibitors, and, in the case of *CDKN2A/MTAP* co-deletions, drugs targeting PRMT5. More generally, our study illustrates how comprehensive precision oncology programs enable the categorization and biological understanding of rare cancers, which in turn can translate back into new approaches to the clinical management of these often difficult-to-treat entities, creating a self-reinforcing loop of forward and reverse translation.

## Methods

### Patients

Patients included in this study had been enrolled in the MASTER[22] or INFORM[23] multicenter precision oncology trials, which use WGS/WES, RNA-seq, and DNA methylation profiling to inform the clinical management of children, adolescents, and young adults with advanced cancers and adults with advanced rare cancers regardless of age. All patients or their legal representatives provided written informed consent for the banking of tumor and control tissue, molecular analysis, the collection of clinical data (including sex and age), and the publication of molecular and clinical information under protocols approved by the Ethics Committee of the Medical Faculty of Heidelberg University (MASTER) or the ethics committee and institutional review board at each participating center (INFORM). The study was conducted in accordance with the Declaration of Helsinki. Patient characteristics are presented in Supplementary Data 1.

### Histopathologic analysis

Immunohistochemical staining of patient tumor sections is described in the Supplementary Methods.

### Genomic and transcriptomic analysis

The processing of tumor and control samples, WGS/WES and RNA-seq, and computational analysis of the resulting data were performed using the standardized workflows of MASTER and INFORM, which are described in detail in the reports on the systematic analysis of the first patient cohorts enrolled in these ongoing programs[22,23]. This also applies to the TFCP2-rearranged sarcomas reported here with the following exceptions: RNA samples from patients TFCP2-HD-6, TFCP2-HD-10, TFCP2-HD-11, and TFCP2-HD-12 were sequenced on an Illumina NovaSeq 6000 instrument with S1 flow cells in paired-end mode (2 × 101 nucleotides [nt]), and WGS samples from patients TFCP2-HD-11 and TFCP2-HD-12 were sequenced on an Illumina NovaSeq 6000 instrument with S4 flow cells in paired-end mode (2 × 151 nt). t-SNE analysis of RNA-seq data was performed with the R package Rtsne version 0.13 on log-transformed expression values of all transcription factors from the Gene Ontology gene set 0000981, except for transcripts that covaried significantly with the date of sequencing since their measurement may be confounded by batch effects. The function removeBatchEffect from the limma package was used to mitigate contamination of the expression profiles of lung metastases by those of surrounding normal lung tissue. Genes differentially expressed between FUS/EWSR1-TFCP2 sarcoma and all other RMS subtypes were calculated using DESeq2 version 1.18.1. KEGG pathway and GOTERM analyses were performed with DAVID (https://david.ncifcrf.gov/tools.jsp)[60].

### DNA methylation analysis

Tumor DNA (250 ng) was analyzed using Infinium MethylationEPIC BeadChip arrays (Illumina). Raw data (idat files) were used to run the sarcoma classifier version 12.2[36]. For further analysis, data were processed using ssNoob from the minfi R package[61]. Unreliable probes, cross-reactive probes, probes mapping to sex chromosomes[62], and probes overlapping with single-nucleotide polymorphisms were filtered out based on dbSNP version 150. Beta values were used for further analyses. All analyses were performed on samples with a sarcoma classifier prediction score >0.9 (except for the 11 TFCP2-rearranged samples) and an estimated tumor cell content of >0.45 based on the leukocyte unmethylation for purity score. Only entities with five or more samples were kept (total number of samples, n = 345). t-SNE clustering was conducted using the M3C and the Rtsne R packages. For entity-based analyses, samples were combined at the probe level, and the mean beta value across all samples was calculated for each entity. Mean beta values were used for hierarchical clustering using Manhattan distance and Ward.D2 linkage. Correlation analysis on mean beta values was performed using Spearman correlation.

### Cell culture, vectors, and lentiviral transduction

Information on the cell lines used, the generation of cDNAs and lentiviral vectors, and the lentiviral transduction of cells is provided in the Supplementary Methods.

### RNA isolation, cDNA synthesis, and quantitative RT-PCR

RNA was isolated using the RNeasy Mini Plus Kit (Qiagen). cDNA synthesis from 1.5 or 2 μg RNA was performed using the High Capacity cDNA RT Kit (Applied Biosystems). Quantitative RT-PCR experiments were run on a C1000 Touch Thermal Cycler (BioRad) with iTAq Universal SYBR Green Supermix (BioRad). Results were analyzed using the $2^{-\Delta\Delta CT}$ or $2^{-\Delta CT}$ method. Primer sequences are provided in Supplementary Table 1.

### Immunoblotting

Protein lysates from cell lines were prepared using RIPA buffer (Merck/Millipore). To prepare protein lysates from patient tumors, 40 to 60 mg of fresh-frozen tumor sections were mixed with 150 μl lysis buffer (8 M urea, 75 mM NaCl, 50 mM Tris pH 8.2, 1 mM $Na_3VO_4$, 1 mM $Na_4P_2O_7$, 1 mM PMSF, 1 mM NaF, 1 mM β-glycerophosphate, 1x Halt protease inhibitor cocktail [Thermo Scientific], 1x EDTA, 0.15 U/μl benzonase [Novagen]) and a stainless steel ball on dry ice and immediately homogenized twice at 5-min intervals using TissueLyser II (Qiagen) at 20 Hz for 2 min. After 30 min on ice, the supernatant was collected after centrifugation at 13,000 revolutions per minute and 4 °C for 30 min. 50 μg protein was used for immunoblotting. SDS-PAGE and western blotting were performed using standard protocols. Detection was performed with HRP-conjugated (Abcam) or near-infrared-labeled (Cell Signaling) secondary antibodies using an Amersham Imager 600 (GE Healthcare) or an Odyssey CLx (LI-COR) system, respectively. Antibodies are provided in the Supplementary Table 2.

### Mass spectrometry

The validation of ALK-ST4 protein expression by mass spectrometry is described in the Supplementary Methods.

### Transformation assays

Colony formation and anchorage-independent growth assays are described in the Supplementary Methods.

## Mouse experiments

Mice were housed in the DKFZ Center for Preclinical Research. All animal procedures were approved by the regional authority in Karlsruhe, Germany (reference number 35-9185.81/G-75/16) and performed according to federal and institutional guidelines. Further experimental details are given in the Supplementary Methods.

## Immunofluorescence and myogenic differentiation analysis

LHCN-M2 cells were seeded in six-well plates at a density of 600,000 per well in normal growth medium. The next day, the medium was changed to differentiation medium containing DMEM/M199 at a 4:1 ratio (Gibco, 10566016 and 31150022), 20 mM Hepes (Biochrom), 0.03 µg/ml zinc sulfate (Sigma-Aldrich), 1.4 µg/ml vitamin B12 (Sigma-Aldrich), 10 µg/ml insulin (Sigma-Aldrich), and 100 µg/ml transferrin (Sigma-Aldrich)[41]. The differentiation medium was replaced daily, and dead cells were removed every other day by gentle rinsing with DPBS before the medium change. Cells were either imaged directly with a Lionheart FX automated microscope or fixed for immunofluorescence to determine the nuclear fusion index. Experimental details on immunofluorescence staining and the determination of the nuclear fusion index are described in the Supplementary Methods.

## ALK inhibitor testing in isogenic MCF10A cells

MCF10A cells stably expressing EV, ALK-WT, ALK-ST1, ALK-ST2, ALK-ST3, or ALK-TPM3 were seeded in white 96-well plates (Corning) at a density of $0.1 \times 10^4$ per well in EGF-depleted medium. The following day, alectinib, ceritinib, or crizotinib (Selleckchem) were added at 10 concentrations ranging from 50 µM to 13 nM with a dilution factor of 2.5. After 72 h, cell viability was determined using CellTiter-Glo Cell Viability Assay (Promega), and luminescence was measured with an EnVision Multimode Microplate Reader (PerkinElmer).

## IC$_{50}$ determination of patient-derived tumor cells

A tumor specimen from patient TFCP2-HD-4 was dissociated into single cells and seeded in 384-well plates at a density of 2000 per well. After 48 h, cells were treated with alectinib, ceritinib, or crizotinib (Hoelzel Diagnostika) in 20 concentrations ranging from 50 µM to 0.1 nM in quadruplicates. After 48 h of incubation, cell viability was assessed with the ATPlite luminescence-based assay (PerkinElmer). Further experimental details and calculation of dose-response curves are described in the Supplementary Methods.

## RNA sequencing of cell lines

MCF10A ($1 \times 10^6$) or SCP-1 ($5 \times 10^6$) cells stably transduced in triplicate (MCF10A) or three to nine replicates (SCP-1) with EV, FUS-TFCP2, EWSR1-TFCP2, FUS, EWSR1, or TFCP2 were seeded in 10- or 15-cm dishes, and RNA was isolated the next day using the RNeasy Mini Plus Kit (Qiagen). Library preparation and 125-nt paired-end read sequencing on an Illumina HiSeq 4000 system were performed at the DKFZ Genomics and Proteomics Core Facility. Details on sequencing, data analysis, and detection of an HRD-associated gene signature in MCF10A and SCP-1 samples are described in the Supplementary Methods.

## Antibody-guided chromatin tagmentation sequencing

MCF10A cells were transduced with two different lentiviral vectors to obtain four biological replicates expressing HA-FUS-TFCP2 or EV (R1–R4) and three biological replicates expressing HA-TFCP2 (R2–R4). For each replicate, ACT-seq was performed with HA-Tag (C29F4) rabbit mAb (Cell Signaling, #3724), HA-Tag (F-7) mouse mAb (Santa Cruz, sc-7392), anti-histone H3 (acetyl K27) antibody−ChIP grade (Abcam, ab4729), and rabbit IgG polyclonal antibody (Merck, #PP64) as negative control. Further details of the ACT-seq procedure and subsequent data analysis are described in the Supplementary Methods.

## Structural predictions

The prediction of motifs and domain structures of ALK-WT and ALK variants is described in the Supplementary Methods.

## Cell viability, apoptosis, and DNA damage analysis

The analysis of cell viability and apoptosis and detection of γH2AX by flow cytometry and immunofluorescence are described in the Supplementary Methods.

## Statistics and reproducibility

Statistical analyses were performed with GraphPad Prism version 8.4.3 and 9.5.1 or with R version 3.4.2. The tests used and the statistics displayed are described in the respective figure legends, and the *p*-values are shown in the graphs. All experiments were performed three times unless specified otherwise. The Benjamini-Hochberg method was used to correct for multiple testing where applicable. No statistical method was used to predetermine the sample size. No data were excluded from the analyses. The experiments were not randomized. The investigators were not blinded to allocation during experiments and outcome assessment.

## Reporting summary

Further information on research design is available in the Nature Portfolio Reporting Summary linked to this article.

## Data availability

Sequencing and DNA methylation data from TFCP2-rearranged patient samples have been deposited in the European Genome-Phenome Archive (EGA) under accession code EGAS00001006939. Sequencing data from Horak et al.[22] are available under accession code EGAS00001004813. All data deposited under accession codes EGAS00001006939 and EGAS00001004813 are available under restricted access [https://ega-archive.org/dacs/EGAC00001000452]. The RNA-seq and ACT-seq data from cell lines generated in this study have been deposited in the Gene Expression Omnibus under accession code GSE224183 and are publicly available. The mass spectrometry data for the detection of ALK-ST4 have been deposited in the ProteomeXchange Consortium via the PRIDE[63] partner repository with the dataset identifier PXD045522 and are publicly available. The deposited data contain raw sequencing, methylation, or mass spectrometry data. Source data including raw values and uncropped western blot scans are provided in this paper.

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

## Acknowledgements

We thank Stefanie Reinhart, Inka Buchroth, Christin Fehmer, and Lydia Fälker for technical assistance; Luisa Schwarzmüller for sharing an important protocol; Barbara Helm for discussing proteomics data analysis and visualization; the NCT/DKFZ Sample Processing Laboratory, the DKFZ Next-Generation Sequencing Core Facility, the DKFZ Omics IT and Data Management Core Facility, and the DKFZ Center for Preclinical Research for technical support. This study was supported by a grant from the German Cancer Aid (70114577) and the NCT 3.0 Integrative Projects in Basic Cancer Research Program. R.I. was supported by the Physician-Scientist Program of Heidelberg University, Faculty of Medicine. A.B. received funding from the European Research Council under the European Union's Horizon 2020 research and innovation program (grant agreement 805338). C.S. was supported by the W2/W3 program for female scientists of the Helmholtz Association (W2_W3-0051). Work in the laboratories of C.R.B., A.B., D.B.L., H.G., D.H., S.F., and C.S. is supported through the HEROES-AYA (Heterogeneity, Evolution, and Resistance in Oncogenic Fusion Gene-Expressing Sarcomas Affecting Adolescents and Young Adults) consortium within the National Decade Against Cancer of the German Federal Ministry of Education and Research (01KD2207). The MASTER program is supported by the NCT Molecular Precision Oncology Program, DKFZ, and DKTK. The INFORM program is supported by DKFZ, DKTK, the German Federal Ministry of Education and Research, the German Federal Ministry of Health, the Ministry of Science, Research, and the Arts of the State of Baden-Württemberg, the German Cancer Aid, the German Childhood Cancer Foundation, RTL television, the aid organization BILD hilft e.V. (Ein Herz für Kinder), and a generous private donation of the Scheu family. G.D.D. and R.B.R. were funded by Wellcome Trust grant 210585/B/18/Z (Impact of missense mutations in recessive Mendelian disease: insight from ciliopathies).

## Author contributions

C.E.H., C.S., and S.F. conceptualized the study; S.U. performed bioinformatic analyses of RNA-seq, WES, WGS, and ACT-seq data; A.M.D. and D.B.L. performed bioinformatic analysis of DNA methylome data; C.E.H., A.N., I.Ø., B.C.J., H.G., W.H., and S.F. participated in patient data collection and curation; W.H. performed histopathology; Dominic H. and M.S. performed proteome analysis; J.S., T.W., and R.I. performed mouse experiments; J.S., K.-S.L., T.W., A.C., T.H., and P.R.-H. performed functional experiments with cell lines; Z.C. and C.R.B. performed drug sensitivity testing with primary patient samples; G.D.D. and R.B.R. performed structural modeling of ALK variants; D.W., M.H., T.K., A.B., and R.I. provided essential methodologies; D.W., C.P., A.N., I.Ø., B.C.J., H.G., and Daniel H. provided essential resources. The original draft was written by C.S. with major input from J.S., S.U., C.E.H., K.-S.L., T.W., A.C., A.M.D., G.D.D., C.R.B., R.B.R., D.B.L., W.H., and S.F. All co-authors reviewed the manuscript.

## Funding

## Competing interests

The authors declare the following competing interests: C.E.H. has received research funding from AstraZeneca, Pfizer, PharmaMar, and Roche. I.Ø. has received funding from AstraZeneca and Pfizer. D.B.L. has received honoraria from Illumina and is an employee of Infectopharm. S.F. has had a consulting or advisory role and received honoraria, research funding, and/or travel/accommodation expenses funding from the following for-profit companies: Amgen, AstraZeneca, Bayer, Eli Lilly, Pfizer, PharmaMar, and Roche. The other authors declare no competing interests.

## Additional information

[1]Division of Applied Functional Genomics, German Cancer Research Center (DKFZ), and National Center for Tumor Diseases (NCT), NCT Heidelberg, a Partnership Between DKFZ and Heidelberg University Hospital, Heidelberg, Germany. [2]Division of Translational Medical Oncology, DKFZ, and NCT Heidelberg, Heidelberg, Germany. [3]Faculty of Biosciences, Heidelberg University, Heidelberg, Germany. [4]Computational Oncology Group, Molecular Precision Oncology Program, NCT Heidelberg, and DKFZ, Heidelberg, Germany. [5]German Cancer Consortium (DKTK), Heidelberg, Germany. [6]Section of Translational Cancer Epigenomics, Division of Translational Medical Oncology, DKFZ, and NCT Heidelberg, Heidelberg, Germany. [7]Division of Cancer Epigenomics, DKFZ, Heidelberg, Germany. [8]Bioquant, Heidelberg University, Heidelberg, Germany. [9]Heidelberg University Biochemistry Center (BZH), Heidelberg, Germany. [10]Department for Translational Medical Oncology, NCT, NCT/UCC Dresden, a Partnership Between DKFZ, Heidelberg Faculty of Medicine and University Hospital Carl Gustav Carus, TUD Dresden University of Technology, and Helmholtz-Zentrum Dresden-Rossendorf (HZDR), Dresden, Germany. [11]German Cancer Consortium (DKTK), Dresden, Germany. [12]Translational Medical Oncology, Faculty of Medicine and University Hospital Carl Gustav Carus, TUD, Dresden, Germany. [13]Faculty of Biology, TUD Dresden University of Technology, Dresden, Germany. [14]University Cancer Center Mainz, Johannes Gutenberg University Mainz, Mainz, Germany. [15]Department of Hematology, Medical Oncology and Pneumology, University Medical Center, Mainz, Germany. [16]German Cancer Consortium (DKTK), Mainz, Germany. [17]Proteomics Core Facility, DKFZ, Heidelberg, Germany. [18]Pediatric Oncology and Coagulation, Karolinska University Hospital, Stockholm, Sweden. [19]Pediatric Oncology and Hematology, Skåne University Hospital, Lund University, Lund, Sweden. [20]Soft-Tissue Sarcoma Junior Research Group, DKFZ, Heidelberg, Germany. [21]Hopp Children's Cancer Center (KiTZ) and NCT Heidelberg, Heidelberg, Germany. [22]Department of Pediatric Oncology, Hematology and Immunology, Heidelberg University Hospital, Heidelberg, Germany. [23]Translational Functional Cancer Genomics, DKFZ, Heidelberg, Germany. [24]Pattern Recognition and Digital Medicine Group, Heidelberg Institute for Stem Cell Technology and Experimental Medicine (HI-STEM), Heidelberg, Germany. [25]Gerhard Domagk Institute of Pathology, University Hospital Münster, Münster, Germany. [26]Institute of Human Genetics, Heidelberg University, Heidelberg, Germany. [27]These authors contributed equally: Julia Schöpf, Sebastian Uhrig, Christoph E. Heilig, Kwang-Seok Lee. [28]These authors jointly supervised this work: Stefan Fröhling, Claudia Scholl. ✉e-mail: stefan.froehling@nct-heidelberg.de; claudia.scholl@nct-heidelberg.de

