## [Peer Review File · Nature Communications]

REVIEWER COMMENTS

Reviewer #1 (Remarks to the Author): Expert in rhabdomyosarcoma functional genomics and therapy

What are the noteworthy results?

This manuscript, submitted by Schopf, Uhrig, Heilig et al, takes advantage of the materials and expertise in two precision oncology programs and deeply analyzes a rare sarcoma called rhabdomyosarcoma with FUS/EWSR1-TFCP2 gene fusions. Analysis consists of multi-omic tools and some functional analyses underlying mechanisms of transformation. The studies are a tour-de-force, and the text does a very good job making the results digestible for the reader, including accurately pointing to figures in the main body of the manuscript and the supplementary materials. The most noteworthy result is that these sarcomas demonstrating skeletal muscle markers do not cluster with classical rhabdomyosarcoma, but with undifferentiated sarcomas. They are a completely different molecular group.

Will the work be of significance to the field and related fields? How does it compare to the established literature? If the work is not original, please provide relevant references.

The work is of significance to the field and demonstrates the power of precision medicine programs. That is, even though the sarcoma community does not yet have many therapeutic options for precision treatment, by deeply analyzing these ultra-rare tumors the community will collectively generate foundational knowledge describing the unanticipated molecular defects, and eventually new medicines will be developed to target these defects. Indeed, other precision medicine programs may look towards this work to guide their own over-arching analytical workflows.

Does the work support the conclusions and claims, or is additional evidence needed?

The provided text, figures, and supplementary material largely support the conclusions and claims, with some exceptions. Also, there could be more “closed loops” so that the reader is not left wondering about some of the observations.

(1) (Line 213) Why is exon 17 of the ALK gene often skipped in these tumors? Can the authors speculate in the Discussion section?

(2) In lines 283-284, the authors conclude that – in contrast to the other ALK variants - the ALK variant ALK-ST4 “had no effect relative to empty vector (EV).” This is difficult to conclude if the variant is not shown to be expressed at the protein level. Specifically, there is no band on the immunoblot in Supp. Figure 3A. Presumably this is because ALK-ST4 is secreted into the extracellular space? If so, the authors need to perform immunoblot analysis of secreted proteins to demonstrate that ALK-ST4 is expressed at the protein level. Otherwise, it is not possible to comment on the functionality of this variant.

(3) Can the authors explain how they chose to focus on ALK, TERT, IGFBP5, and PAPP2 to investigate or think about? There were so many other genes with even more significant changes higher on the list.

(4) Did the authors note any correlations/associations between age of the patient from which the tumors were obtained and any genetic/genomic findings?

(5) While the observations on TERT are interesting, they are unfinished. It is not reasonable to perform a whole new study on the functionality of these TERT changes – indeed the manuscript is long enough - but at minimum the authors should try to perform an immunoblot to show that the TERT proteins encoded by these truncated transcripts are expressed at the protein level.

Are there any flaws in the data analysis, interpretation and conclusions? Do these prohibit publication or require revision?

To this reviewer, there is some over-generalization and over-interpretation in the text.

(6) In the introduction (line 95) the authors state that RMS “...is composed of malignant immature precursor cells with myogenic differentiation...” It would be more appropriate to state that there is “partial myogenic differentiation.”

(7) In the Figure 1 legend, the authors state in (A) that there was a “malignant relapse.” But one cannot state that it was a relapse if it was not malignant to begin with. It might work better to state that there was a recurrence. In (B) the authors use the term “rhabdoid” cells. Are they referring to rhabdoid tumors? Or are they referring to cells with skeletal muscle histogenesis? It is important to clarify this, since some readers may make the leap to SMARCB1 mutated rhabdoid tumors, even if this is not intended.

(8) In lines 295-297, the authors state that they "...failed to produce SCP-1 cells with stable overexpression of ALK-ST2 since they always died shortly after transduction, presumably due to oncogenic stress." Why wouldn't ALK-ST1 have a similar toxic effect, since it is almost identical to ALK-ST2 including retaining the kinase domain? Please address this in the text.

(9) In lines 545-546 the authors provide some thoughts on the possible impact of the TERT variants on tumorigenesis by pointing to a publication (reference 54). Please expand a little on this. For example, do the non-canonical effects of TERT such as effects on proliferation and invasion require the full-length TERT transcript? If they do, then this hypothesis cannot stand.

Is the methodology sound? Does the work meet the expected standards in your field?

(10) it is important to acknowledge that MCF10A cells, while having some practical benefits for study, also have some limitations, since they are not mesenchymal and do not have an epigenetic/epigenomic background mirroring the cells of origin of FUS/EWSR1-TFCP2 tumors. [It is understood that we don't know the cell of origin, and that is one reason why the authors also used SCP-1 cells for their studies (mesenchymal cell line).]

(11) What is the functional status of p53 in these FUS/EWSR1-TFCP2 tumors? How should we interpret the DNA damage studies given the MCF10A cells are p53-deficient? Additional wet lab experiments are not necessary, but it is reasonable for the authors to apply p53 deficient gene signatures to these tumors to predict in silico whether p53 tumor suppressor function is deficient.

(12) Why were the xenograft experiments terminated when tumor volume reached 500mm³ (Fig.3C)? What would happen if the tumor volume were to get closer to 1500m³, which is a more typical endpoint for xenograft studies?

Is there enough detail provided in the methods for the work to be reproduced?

There is enough detail provided in the methods for the work to be reproduced.

Reviewer #2 (Remarks to the Author): Expert in rhabdomyosarcoma genomics, functional genomics, epigenomics, and therapy

In their manuscript “Multi-Omic and Functional Analysis for Classification and Therapeutic Targeting of Ultra-Rare Soft-Tissue Sarcomas with FUS-TFCP2 or EWSR1-TFCP2 Fusions” Schöpf et al. describe cases of very rare soft tissue sarcomas expressing FUS-TFCP2 or EWSR1-TFCP2 fusion genes. They find that these tumors express high levels of ALK and new ALK variants, which seem to be oncogenic according to in vitro assays performed by the authors. The authors describe that these sarcomas have highly rearranged genomes in spite of having little recurrent gene mutations. Their data suggests that the fusion genes themselves may contribute to the genomic instability observed. Overall, this is a well written and potentially important report based on data from two large precision oncology programs. The information provided here about these rare tumors is relevant to the field, but unfortunately lacks mechanistic detail. Importantly, the in vitro data is, as far as I understood this correctly, in stark contrast to clinical observations, which have shown that these tumors are resistant to both genotoxic and ALK inhibitor therapy. The authors argue that this may be due to the ALK variants expressed in patients, but do not explore this hypothesis in detail, reducing the clinical impact of their observations. If DNA damage repair was impaired and ALK was a driving event, one would expect at least a subset of patients to respond to genotoxic treatment regimens and ALK inhibitor treatment.

Additional major comments:

1. The patient characteristics are mentioned in the text, but no summary of these characteristics is presented in a figure or a table. I would suggest adding such a table/figure summarizing the clinical features.
2. The authors clearly show that ectopic expression of the ALK variants can lead to oncogenic properties in cells and that this increases their sensitivity to ALK inhibitors. Ideally, one would also show that the expression of these variants is required to maintain the transformed phenotype by LOF experiments, eg. knock down. Do the authors have the possibility to perform such assays in fusion-expressing cell line models? What do they observe?
3. It remains unclear which cell lines were derived from which patient and how well their response to ALK inhibitors or genotoxic treatment in vitro is associated with the response observed in the patients these cells were derived from. This is an essential piece of information that would help the reader interpret the results of these experiments.
4. In general, it would be important to understand if high ALK expression or ALK variant expression drives high signaling in these cells. Unfortunately, no evidence is provided about the activation of ALK signaling.
5. The authors demonstrate that the fusion leads to defects in myogenic differentiation, but do not provide mechanistic insights of how this happens. The ACT seq, RNAseq and ChIP seq should be useful to address this question. Can the authors identify pathways that are related to disrupted muscle

differentiation there that are directly dysregulated by the fusion through their binding to genes that would promote de-differentiation?

6. The ACT-seq experiments do not show very strong signals/ peaks. I am not an expert in this method, but I am not sure if the peaks shown in the main figure really support binding to ALK for example. Could the authors quantify the peak strength and/or rank the peaks and show where ALK and TERT rank compare to other peaks? Also, the ChIPseq data for some active chromatin marks eg H3K27Ac does not seem to change after fusion expression. It would be ideal to compare the binding of the fusion to the degree of chromatin activation. Does the fusion really increase chromatin activation?

7. It remains unclear how the fusion influences DNA damage repair. It does seem that expression of the fusion is sufficient to increase DNA damage, but it is unclear what causes this DNA damage, eg. could it be fusion-induced replication stress as observed for EWS-FLI1 in Ewing's sarcomas?

Reviewer #3 (Remarks to the Author): Expert in protein structure prediction and structural biology

I was asked to review especially the structural modelling part of the work of Schöpf et al. thus I limit my assessment exclusively on this part. For the use the authors are doing of the structural model produced for the human ALK protein I think the modelling is well done and sufficient to interpret their data, as they limit their analysis only to structured domains whose homology is well captured by HHpred. A more comprehensive model is provided also at the AlphaFold database (<https://alphafold.ebi.ac.uk/entry/Q9UM73>) which the authors can also use and/or compare with their model. Note that in the AF model, while the single domains are well predicted, likely with similar accuracy as HHpred, their reciprocal location is uncertain (see the large expected position error), thus the comparative analysis might have some meaning only for the single domains.

REVIEWER COMMENTS

Reviewer #1 (Remarks to the Author): Expert in rhabdomyosarcoma functional genomics and therapy

What are the noteworthy results?

This manuscript, submitted by Schopf, Uhrig, Heilig et al, takes advantage of the materials and expertise in two precision oncology programs and deeply analyzes a rare sarcoma called rhabdomyosarcoma with FUS/EWSR1-TFCP2 gene fusions. Analysis consists of multi-omic tools and some functional analyses underlying mechanisms of transformation. The studies are a tour-de-force, and the text does a very good job making the results digestible for the reader, including accurately pointing to figures in the main body of the manuscript and the supplementary materials. The most noteworthy result is that these sarcomas demonstrating skeletal muscle markers do not cluster with classical rhabdomyosarcoma, but with undifferentiated sarcomas. They are a completely different molecular group.

Will the work be of significance to the field and related fields? How does it compare to the established literature? If the work is not original, please provide relevant references.

The work is of significance to the field and demonstrates the power of precision medicine programs. That is, even though the sarcoma community does not yet have many therapeutic options for precision treatment, by deeply analyzing these ultra-rare tumors the community will collectively generate foundational knowledge describing the unanticipated molecular defects, and eventually new medicines will be developed to target these defects. Indeed, other precision medicine programs may look towards this work to guide their own over-arching analytical workflows.

Does the work support the conclusions and claims, or is additional evidence needed?

The provided text, figures, and supplementary material largely support the conclusions and claims, with some exceptions. Also, there could be more “closed loops” so that the reader is not left wondering about some of the observations.

Response: Thank you for carefully reviewing our manuscript, the positive and encouraging comments, and the constructive feedback. We addressed the concerns raised with further experiments and adjusted the manuscript text to improve clarity. Changes are highlighted in red in the revised manuscript and referenced by line numbers in our point-by-point response. In addition, several changes were made to comply with the format of *Nature Communications*, e.g., shortening the title and abstract.

(1) (Line 213) Why is exon 17 of the ALK gene often skipped in these tumors? Can the authors speculate in the Discussion section?

Response: This is indeed an interesting observation, and we explored several possible explanations for the preferential skipping of ALK exon 17:

First, we assessed whether the intragenic ALK deletions eliminated the splice acceptor site of exon 17. However, they always started in intron 16 upstream of exon 17 (ALK is transcribed from the reverse strand), so the splice acceptor site was not affected.

Second, we examined whether the splice acceptor motifs of exons 17 and 18 differed from each other and whether only that of exon 18 matched the splice donor motif of exon 2. The

rationale for this analysis was that exons 2 and 18 were the most frequent splice donor and acceptor, respectively, in the tumor samples, and a better match between the splice site motifs of exon 2 and 18 could explain the preferential skipping of exon 17. However, both exons 17 and 18 followed the canonical splice motif GU-AG, and exon 17 even had the more frequent CAG splice acceptor site.

Third, we considered whether a potential splice junction between exons 2 and 17 would result in a frameshift that generates a transcript subject to nonsense-mediated decay. This scenario could force tumor cells to skip exon 17 to generate an in-frame splice junction between exons 2 and 18 to express oncogenic *ALK*. However, splicing between exons 2 and 17 was also in-frame.

Thus, the preferential skipping of *ALK* exon 17 remains unexplained at this time. Considering that the manuscript already exceeds the permitted word count, we have not included these hypotheses and analyses in the revised manuscript but would be happy to do so if found sufficiently important by the reviewer and editor.

(2) In lines 283-284, the authors conclude that – in contrast to the other ALK variants - the ALK variant ALK-ST4 “had no effect relative to empty vector (EV).” This is difficult to conclude if the variant is not shown to be expressed at the protein level. Specifically, there is no band on the immunoblot in Supp. Figure 3A. Presumably this is because ALK-ST4 is secreted into the extracellular space? If so, the authors need to perform immunoblot analysis of secreted proteins to demonstrate that ALK-ST4 is expressed at the protein level. Otherwise, it is not possible to comment on the functionality of this variant.

Response: We fully agree. There is no protein band in **Supplementary Fig. 3a** because the antibody binds to an ALK portion that is lost in ALK-ST4. Unfortunately, we could not find a working antibody that binds the N-terminus of ALK, and thus ALK-ST4, and therefore only confirmed ALK-ST4 mRNA expression by qRT-PCR, as mentioned in the legend to **Supplementary Fig. 3a**. To definitively demonstrate ALK-ST4 protein expression, we now performed mass spectrometry-based label-free quantitative proteomics on lysates from MCF10A cells stably expressing empty vector (EV), wildtype ALK (ALK-WT), or ALK-ST4. Peptides/proteins were identified with DIA-NN version 1.8.1 (Demichev et al. 2020) using a UniProt reference database (downloaded on July 18, 2023) containing the ALK-WT sequence and complemented with the ALK-ST4 amino acid sequence as an individual entry. To detect and distinguish ALK-WT and ALK-ST4 despite the lack of unique ALK-ST4 peptides, we annotated ALK-WT-specific peptides as “ALK-WT” (**Response Fig. 1a**, in purple) and the common peptides between ALK-WT and ALK-ST4 as “ALK-ST4” (**Response Fig. 1a**, in red). To compare protein abundance, we collapsed peptide intensities to the protein level by summing peptide intensity per protein name, followed by quantile normalization across all samples. As a result, we identified “ALK-WT”-specific peptides only in cells expressing ALK-WT, whereas we did not detect these peptides in cells transduced with EV or ALK-ST4, as expected (**Response Fig. 1b**). In contrast, we identified the overlapping “ALK-ST4” peptides in cells expressing ALK-WT and ALK-ST4 but not in EV cells (**Response Fig. 1b**). Taken together, ALK-ST4 protein is expressed to an at least similar level as ALK-WT, and we therefore feel confident to conclude that it has no oncogenic potential.

In addition, we performed the same proteomics analysis on 1 ml of conditioned medium after two-day culture of EV- and ALK-ST4-transduced MCF10A cells. However, we did not detect ALK peptides in the cell supernatant, indicating that ALK-ST4 is not secreted.

We added this new information to the legend of **Supplementary Fig. 3a**: “*ALK-ST4 cannot be detected with the antibody used as it binds to the intracellular part of ALK, but mRNA and protein expression were confirmed by qRT-PCR and mass spectrometry-based label-free quantitative proteomics, respectively (data not shown).*” We also added the following statement to the manuscript text (line 268): “(Supplementary Fig. 3a; *ALK-ST4 protein expression was confirmed by mass spectrometry [data not shown].*)”

a

b

Response Fig. 1: Confirmation of ALK-ST4 protein expression in MCF10A cells. **(a)** Mapping of peptides identified by proteomics (colored) to the amino acid sequences of ALK-WT and ALK-ST4. Shared peptides are colored red. ALK-WT-specific peptides are colored purple. **(b)** Quantitative comparison of ALK-WT and ALK-ST4 protein expression. Four technical replicates were used. Bio1 and Bio2 represent two independently generated cell lines, and Bio1 cells were used for the functional experiments presented in the manuscript. Shown are the sum of peptide intensity per protein and log₂-transformed values. 0 (zero) indicates that proteins were not detected or below the detection limit.

Reference:

Demichev et al. *DIA-NN: Neural networks and interference correction enable deep proteome coverage in high throughput.* *Nat Methods* 17:41–44 (2020)

(3) Can the authors explain how they chose to focus on ALK, TERT, IGFBP5, and PAPP2 to investigate or think about? There were so many other genes with even more significant changes higher on the list.

Response: We are happy to explain why we selected ALK, TERT, IGFBP5, and PAPP2. The project's starting point was an analysis of RNA-seq data from all patients enrolled in the MASTER precision oncology program to identify potential drug targets. This revealed extreme overexpression of ALK in a subset of patients who all harbored FUS/EWSR1-TFCP2 fusions, which had not been reported then. Based on this discovery, we initially focused on ALK and identified several abnormalities, followed by detailed mechanistic studies. As introducing TFCP2 fusions into immortalized cells caused transcriptional upregulation of ALK, we performed ACT-seq to determine that this was a direct effect. Upon close inspection of the ACT-seq data, we also noticed very prominent binding of FUS-TFCP2 and TFCP2 upstream of TERT exon 3. We then interrogated the patients' RNA-seq data and found that this binding translated into extremely high TERT expression compared to other tumor entities. Given the relevance of TERT for cancer development, we felt that this was worth reporting, although the function of the truncated TERT variant remains to be investigated.

We also reported IGFBP5 because it was the most upregulated gene after FUS-TFCP2 and EWSR1-TFCP2 expression in both MCF10A and SCP-1 cells (**Figure 5c**) and, at the same time, was extremely highly expressed in TFCP2-rearranged patient samples compared with all other tumor entities (**Figure 5d**) and with other RMS cases (**Supplementary Fig. 2H**). Upon closer examination of the function of the few genes recurrently mutated in TFCP2-rearranged sarcomas, we found that PAPP2 is an essential regulator of IGFBP5. This link and previous reports of a functional role for IGFBP5 in muscle cells strengthened our hypothesis that deregulation of IGFBP5 may play a role in the pathogenesis of this disease.

We hope these explanations and the revised manuscript clarify why we prioritized studying the genes mentioned above as candidate drivers of TFCP-rearranged sarcomas.

(4) Did the authors note any correlations/associations between age of the patient from which the tumors were obtained and any genetic/genomic findings?

Response: Given our cohort's wide age range, it is indeed interesting to search for associations between molecular alterations and early or late disease onset.

The following genomic and transcriptomic characteristics showed no statistically significant correlation with patient age at diagnosis: Fusion partner of TFCP2 (FUS versus EWSR1), CDKN2A loss, intragenic ALK deletion, average homologous recombination deficiency (HRD) score, and expression levels of ALK, IGFBP5, TERT, and CCND2. There were trends toward positive correlations between advanced age and high ALK and IGFBP5 expression (Spearman's rank correlation coefficients of 0.35 and 0.31, respectively, and p-values of 0.072 and 0.095, respectively) and a trend toward a negative correlation between advanced age and high TERT expression (Spearman's rank correlation coefficient of -0.39 and p-value of 0.059).

We also searched for rare germline mutations in cancer predisposition genes, but there were no recurrent variants that would explain the early versus late onset of disease.

(5) While the observations on TERT are interesting, they are unfinished. It is not reasonable to perform a whole new study on the functionality of these TERT changes – indeed the manuscript is long enough - but at minimum the authors should try to perform an immunoblot to show that the TERT proteins encoded by these truncated transcripts are expressed at the protein level.

Response: Thank you for this important suggestion. From five patients, we had fresh-frozen tumor tissue for detecting the TERT Δ ex1-2 variant by western blotting using an antibody against the C-terminus of TERT. As can be seen from the new **Fig. 5i**, we detected high levels of a shortened TERT protein of around 50 kDa in all tumor samples but not in HeLa control cells. These data are described in the Results (line 427) and Discussion (line 556) sections.

Are there any flaws in the data analysis, interpretation and conclusions? Do these prohibit publication or require revision?

To this reviewer, there is some over-generalization and over-interpretation in the text.

(6) In the introduction (line 95) the authors state that RMS “...is composed of malignant immature precursor cells with myogenic differentiation...” It would be more appropriate to state that there is “partial myogenic differentiation.”

Response: Thank you for bringing this to our attention. We have amended the text accordingly, which now reads as follows (line 81): “*Rhabdomyosarcoma (RMS) is a STS subtype composed of malignant immature precursor cells with partial myogenic differentiation defined by aberrant expression of the myogenic transcription factors MYOD1 and MYOG.*”

(7) In the Figure 1 legend, the authors state in (A) that there was a “malignant relapse.” But once cannot state that it was a relapse if it was not malignant to begin with. It might work better to state that there was a recurrence. In (B) the authors use the term “rhabdoid” cells. Are they referring to rhabdoid tumors? Or are they referring to cells with skeletal muscle histogenesis? It is important to clarify this, since some readers may make the leap to SMARCB1 mutated rhabdoid tumors, even if this is not intended.

Response: We agree that the potential transition of a benign lesion into a RMS can be described more precisely and have changed “*Malignant relapse*” to “*Recurrence with histopathologic criteria of malignancy*” in the legend to Fig. 1a and “*Relapse*” to “*Recurrence*” in Fig. 1a.

We have used “rhabdoid” to describe the tumor cells which, partially, showed abundant cytoplasm and eccentric nuclei typical of rhabdoid morphology. The term is commonly used in the workup of tumor specimens in surgical pathology and does not refer to rhabdoid tumors as a diagnostic category in our work. It has also been used previously in describing TFCP2-rearranged RMS (e.g. Loarer et al. 2020).

Reference:

Loarer et al. A subset of epithelioid and spindle cell rhabdomyosarcomas is associated with TFCP2 fusions and common ALK upregulation. Mod Pathol 33:404–419 (2020)

(8) In lines 295-297, the authors state that they "...failed to produce SCP-1 cells with stable overexpression of ALK-ST2 since they always died shortly after transduction, presumably due to oncogenic stress." Why wouldn't ALK-ST1 have a similar toxic effect, since it is almost identical to ALK-ST2 including retaining the kinase domain? Please address this in the text.

Response: Our experience with SCP-1 cells is that they cope much worse with stress than MCF10A cells, which is why the latter are often used to analyze highly transforming oncogenes. We transduced SCP-1 cells many times over more than a year with lentiviral ALK-ST2, using the same viruses that we have successfully used for MCF10A, but they died each time. Also, after lentiviral transduction with ALK-ST1, SCP-1 cells needed more time to adapt but eventually did. The statement that we failed to produce SCP-1 cells with ALK-ST2 due to oncogenic stress was based on the observation that ALK-ST2 exhibited stronger oncogenicity than ALK-ST1 in MCF10A cells. Despite similar expression levels (**Supplementary Fig. 3a**), ALK-ST2 induced higher proliferation (**Fig. 3a**), more colonies in soft agar (**Fig. 3b**), and faster tumor formation in mice (**Fig. 3c**) compared with ALK-ST1. In addition, we now provide new data on the activation of signaling pathways downstream of ALK by the newly discovered ALK variants. Using immunoblotting, we found that ALK-ST2 most strongly activates downstream signaling, i.e., the MEK-ERK and PI3K-AKT pathways, compared with ALK-ST1 and ALK-ST3 in MCF10A cells (**Supplementary Fig. 3C**), further supporting its stronger oncogenicity.

We have added the following sentence to the manuscript to be more clear on this point (lines 290-293): "*We failed to produce SCP-1 cells with stable overexpression of ALK-ST2 since they always died shortly after transduction, presumably due to oncogenic stress, as ALK-ST2 exhibited the strongest oncogenicity and pathway activation in MCF10A cells (Fig. 3a-c; Supplementary Fig. 3c).*"

(9) In lines 545-546 the authors provide some thoughts on the possible impact of the TERT variants on tumorigenesis by pointing to a publication (reference 54). Please expand a little on this. For example, do the non-canonical effects of TERT such as effects on proliferation and invasion require the full-length TERT transcript? If they do, then this hypothesis cannot stand.

Response: This is a good point that indeed needs further clarification. Reference 54 (Pestana et al. 2017) is a comprehensive review on non-canonical TERT functions. It covers mainly full-length TERT, but also includes a few studies on TERT splice variants. For full-length TERT, multiple studies have shown that it affects cell proliferation and invasion independently of its telomere maintenance function. One study demonstrated that the overexpression of the TERT splice variant $\Delta 4-13$, which lacks exons 4 to 13 encoding the telomerase catalytic domain, resulted in increased proliferation of several cell types without affecting telomerase activity (Hrdlickova et al. 2012). However, this variant lacks different regions than our $\Delta ex1-2$ variant, so a possible telomerase-independent function of TERT $\Delta ex1-2$ on proliferation remains speculative. We have amended the Discussion section accordingly (lines 560-563).

In addition, TERT is involved in DNA repair independent of its canonical immortalization function (Pestana et al. 2017). For example, as outlined in the Discussion section, cells with persistent TERT loss were shown to exhibit altered overall chromatin structure, impaired activation of the DNA damage response, enhanced radiosensitivity, and increased numbers of chromosome fragments upon irradiation (Masutomi et al. 2005). Thus, expression of the non-functional $\Delta ex1-2$ variant could phenocopy TERT loss and, therefore, impair DNA repair.

References:

Hrdlickova et al. Alternatively spliced telomerase reverse transcriptase variants lacking telomerase activity stimulate cell proliferation. *Mol Cell Biol* 32:4283–4296 (2012)

Masutomi et al. The telomerase reverse transcriptase regulates chromatin state and DNA damage responses. *Proc National Acad Sci* 102, 8222–8227 (2005)

Pestana et al. TERT biology and function in cancer: beyond immortalisation. *J Mol Endocrinol* 58, R129–R146 (2017)

Is the methodology sound? Does the work meet the expected standards in your field?

(10) *it is important to acknowledge that MCF10A cells, while having some practical benefits for study, also have some limitations, since they are not mesenchymal and do not have an epigenetic/epigenomic background mirroring the cells of origin of FUS/EWSR1-TFCP2 tumors. [It is understood that we don't know the cell of origin, and that is one reason why the authors also used SCP-1 cells for their studies (mesenchymal cell line).]*

Response: We fully agree and have changed the text to better convey this important point (lines 285-288). The respective sentence now reads: “*Since RMS is a mesenchymal malignancy and MCF10A cells are of epithelial origin, we validated these results in the immortalized mesenchymal stem cell line SCP-1, which may better reflect the epigenomic state of the yet unknown cell of origin of FUS/EWSR1-TFCP2 tumors (Supplementary Fig. 3f).*”

(11) *What is the functional status of p53 in these FUS/EWSR1-TFCP2 tumors? How should we interpret the DNA damage studies given the MCF10A cells are p53-deficient? Additional wet lab experiments are not necessary, but it is reasonable for the authors to apply p53 deficient gene signatures to these tumors to predict in silico whether p53 tumor suppressor function is deficient.*

Response: This is an interesting point that we addressed with the proposed analyses. The TFCP2-rearranged tumors showed no *TP53* genomic alterations. Furthermore, they grouped with *TP53*-wildtype sarcomas from the MASTER study (Horak et al. 2021) in a hierarchical cluster analysis with RNA-seq data of KEGG pathway “p53 signaling” genes (hsa04115; **Response Fig. 2**). Taken together, these analyses demonstrate that FUS/EWSR1-TFCP2-positive tumors are TP53-proficient.

Response Fig. 2: Hierarchical clustering of KEGG pathway “p53 signaling” genes in RNA-seq data from the FUS/EWSR1-TFCP2 cases (TFCP2_RMS) and sarcomas enrolled in the MASTER program until November 21, 2018. N, no; Y, yes; mut, mutant.

How and to what extent p53 deficiency in MCF10A cells affects the results of our DNA damage studies is difficult to assess. It is known that p53 contributes to genome maintenance by arresting the cell cycle in the presence of DNA damage to allow DNA repair (Williams & Schumacher 2016). Recent studies showed that p53 also directly impacts the activity of various DNA repair systems. Still, it remains challenging to mechanistically decipher the various functions of p53 in this regard.

We investigated the effects of FUS-TFCP2 on apoptosis and γ H2AX levels by comparing them with those of wildtype FUS, wildtype TFCP2, and EV expressed on the same p53-deficient background. We are, therefore, confident that the observed increase in cisplatin-induced apoptosis and γ H2AX levels is due to FUS-TFCP2, although it is possible that the magnitude of effects would be different in p53-proficient cells. Finally, we observed the enrichment of an HRD-associated gene signature in response to the expression of FUS-TFCP2 or EWSR1-TFCP2 compared with FUS, TFCP2, and EV in both p53-deficient MCF10A and p53-proficient SCP-1 cells (**Fig. 6a**), further supporting the notion that TFCP2 fusions impact the DNA repair machinery independently of p53 status. The p53 wildtype status of SCP-1 cells has been reported previously (Böcker et al. 2008) and was confirmed in our study using RNA-seq.

References:

Böcker et al. Introducing a single-cell-derived human mesenchymal stem cell line expressing hTERT after lentiviral gene transfer. J Cell Mol Med 12:1347–1359 (2008)

Horak et al. Comprehensive genomic and transcriptomic analysis for guiding therapeutic decisions in patients with rare cancers. Cancer Discov 11:2780–2795 (2021)

Williams & Schumacher. p53 in the DNA-damage-repair process. Cold Spring Harb Perspect Med 6:a026070 (2016)

(12) *Why were the xenograft experiments terminated when tumor volume reached 500mm³ (Fig.3C)? What would happen if the tumor volume were to get closer to 1500m³, which is a more typical endpoint for xenograft studies?*

Response: Thank you for this question, which made us realize that we need to adequately explain this experiment. Each cell line was applied to both flanks of three mice, resulting in six tumors per group. Tumor width and length were measured with a caliper, and tumor volume

was calculated using the formula $(\text{length} \times \text{width} \times \text{width})/2$. After reaching the maximum allowed tumor length of 1.5 cm, which is below a volume of 1,500 mm³ due to a more ellipsoid form of the tumors, the mice had to be sacrificed even if the tumor of the other flank had not yet reached the maximum allowed size. Thus, we have in each group three tumors that are much smaller than the maximum size. This was not well explained in the Methods section, and we have amended the Supplementary Methods (page 4) as follows: “Each cell line was applied to both flanks of three mice. After injection, animals were monitored closely, and the width and length of the tumor was measured with a caliper. Tumor volumes were calculated according to the formula $(\text{length} \times \text{width} \times \text{width})/2$. After reaching the maximum allowed tumor length of 1.5 cm, mice were sacrificed even if the tumor of the other flank had not yet reached the maximum allowed size.”

In addition, we apologize that the data presented in **Fig. 3c** were not explained well. This graph displays the tumor volumes per cell line until the first animal had to be sacrificed, so each data point shows all six tumors per group. Otherwise, subsequent time points would have only four or two tumors, making interpretation of the plot difficult. However, the remaining animals were followed and not sacrificed until one of their tumors reached the maximum length. One mouse had to be sacrificed before the maximum tumor size was reached because it had an open wound in the tumor area. The progression of each individual tumor is shown in **Response Fig. 3**. To better explain the data displayed in **Fig. 3c**, we added the following sentence to the legend (lines 1010-1012): “Shown is the mean volume \pm SEM of six tumors per cell line until the first mouse had to be sacrificed in one group due to reaching the maximum allowed tumor length.”

Response Fig. 3: Tumor growth in NOD-SCID mice of MCF10A cells stably transduced with the newly discovered oncogenic ALK alterations. Shown is the tumor growth of each individual tumor over time. The two tumors marked with an asterisk (*) were from the mouse that had to be sacrificed before the tumors reached the maximum allowable size due to an open wound in the tumor area.

Is there enough detail provided in the methods for the work to be reproduced?

There is enough detail provided in the methods for the work to be reproduced.

Reviewer #2 (Remarks to the Author): Expert in rhabdomyosarcoma genomics, functional genomics, epigenomics, and therapy

In their manuscript “Multi-Omic and Functional Analysis for Classification and Therapeutic Targeting of Ultra-Rare Soft-Tissue Sarcomas with FUS-TFCP2 or EWSR1-TFCP2 Fusions” Schöpf et al. describe cases of very rare soft tissue sarcomas expressing FUS-TFCP2 or EWSR1-TFCP2 fusion genes. They find that these tumors express high levels of ALK and new ALK variants, which seem to be oncogenic according to in vitro assays performed by the authors. The authors describe that these sarcomas have highly rearranged genomes in spite of having little recurrent gene mutations. Their data suggests that the fusion genes themselves may contribute to the genomic instability observed. Overall, this is a well written and potentially important report based on data from two large precision oncology programs. The information provided here about these rare tumors is relevant to the field, but unfortunately lacks mechanistic detail. Importantly, the in vitro data is, as far as I understood this correctly, in stark contrast to clinical observations, which have shown that these tumors are resistant to both genotoxic and ALK inhibitor therapy. The authors argue that this may be due to the ALK variants expressed in patients, but do not explore this hypothesis in detail, reducing the clinical impact of their observations. If DNA damage repair was impaired and ALK was a driving event, one would expect at least a subset of patients to respond to genotoxic treatment regimens and ALK inhibitor treatment.

Response: Thank you for carefully reviewing our manuscript and the constructive feedback. We addressed the concerns raised with further experiments and adjusted the manuscript text to improve clarity. Changes are highlighted in red in the revised manuscript and referenced by line numbers in our point-by-point response. In addition, several changes were made to comply with the format of *Nature Communications*, e.g., shortening the title and abstract.

We are grateful for raising the important question to what extent our *in vitro* data allow conclusions about the efficacy of HRD-directed agents and ALK inhibitors in patients with TFCP2-rearranged sarcomas, which we address here and in more detail in the specific comments below.

Molecular footprints of impaired DNA repair, such as elevated HRD scores, predict clinical benefit from platinum-based chemotherapy and PARP inhibitors in epithelial cancers (Coleman et al. 2017, Mirza et al. 2016, Telli et al. 2016). However, it remains to be determined if this also applies to mesenchymal cancers.

In our cohort, three patients received platinum-based chemotherapy (cisplatin, n=1; carboplatin, n=2), and two patients received the PARP inhibitor olaparib combined with other chemotherapeutics (**Supplementary Table 1**). The two patients treated with carboplatin received this drug as part of second-line (TFCP2-HD-3) and third-line (TFCP2-HD-7) combinations following exposure to other chemotherapy regimens and did not achieve an objective response. Patient TFCP2-HD-10 received neoadjuvant cisplatin in combination with docetaxel and 5-fluorouracil, followed by surgery and adjuvant irradiation during primary treatment and had progressive disease after seven months. In the two patients treated with olaparib-containing regimens, disease stabilization was achieved, in contrast to the results of previous therapies. In patient TFCP2-HD-8, salvage surgery was performed due to progressive disease during neoadjuvant ifosfamide, vincristine, and actinomycin. After the next progression, treatment with olaparib, vincristine, irinotecan, and temozolomide resulted in disease stabilization for seven months. In patient TFCP2-HD-5, who progressed immediately after completion of adjuvant chemotherapy with doxorubicin and ifosfamide,

olaparib combined with trabectedin stabilized the disease for four months. Our interpretation of these data is that HRD-targeted therapy, specifically PARP inhibition, has activity in extensively pretreated TFCP2-rearranged sarcomas refractory to standard chemotherapy. In our view, the observed disease stabilizations, which are very unusual in advanced-stage TFCP2-rearranged sarcomas, represent an important signal and warrant testing PARP inhibitors in earlier treatment lines when objective remissions are more likely than in patients who have exhausted multiple conventional therapies. Considering that HRD-deficient carcinomas are also sensitive to platinum derivatives, we believe that the role of these agents, most likely as part of rational combination therapies, in the management of newly diagnosed TFCP2-rearranged sarcomas should also be evaluated to detect potential signs of activity masked in late-stage multidrug-resistant tumors.

Concerning ALK inhibition, we now present new experimental data showing that the novel oncogenic variants ALK-ST1, ALK-ST2, and ALK-ST3 have differential sensitivity to crizotinib, ceritinib, and alectinib and that these inhibitors have varying efficacy against the ALK variants, with ceritinib being the most effective, followed by crizotinib and alectinib (**Fig. 3d**; please also see our response to comment 2 below). We agree that the clinical courses of our patients are very unfavorable. However, most patients were treated with crizotinib, and the only patient (TFCP-HD-4) who received the, according to our *in vitro* experiments, most effective drug, ceritinib, started treatment very late when his overall condition was already so compromised that response could not be assessed (please also see our response to comment 3 below).

We hope the reviewer agrees that our discovery of HRD and oncogenic ALK variants in TFCP2-rearranged sarcomas provides the basis for future investigations into the value of these molecular alterations as therapeutic targets in an appropriate clinical setting. This should include testing HRD-targeted approaches and ALK inhibition in early lines of treatment instead of end-stage disease, as implemented in other indications such as mutant kinase-driven lung cancer, as well as rational combinations rather than single-agent therapies.

In the revised manuscript (lines 526-533), we discuss the concern that evaluating ALK inhibitors as monotherapy in far-advanced tumors that have undergone extensive evolution and selection of aggressive cell populations may lead to premature dismissal of a valid treatment option urgently needed in an entity without effective medical therapies.

References:

Coleman et al. Rucaparib maintenance treatment for recurrent ovarian carcinoma after response to platinum therapy (ARIEL3): a randomised, double-blind, placebo-controlled, phase 3 trial. Lancet 390: 1949–1961 (2017)

Mirza et al. Niraparib maintenance therapy in platinum-sensitive, recurrent ovarian cancer. N Engl J Med 375:2154–2164 (2016)

Telli et al. Homologous recombination deficiency (HRD) score predicts response to platinum-containing neoadjuvant chemotherapy in patients with triple-negative breast Cancer. Clin Cancer Res 22:3764–3773 (2016)

Additional major comments:

1. *The patient characteristics are mentioned in the text, but no summary of these characteristics is presented in a figure or a table. I would suggest adding such a table/figure summarizing the clinical features.*

Response: Detailed patient characteristics are provided in **Supplementary Table 1**.

2. *The authors clearly show that ectopic expression of the ALK variants can lead to oncogenic properties in cells and that this increases their sensitivity to ALK inhibitors. Ideally, one would also show that the expression of these variants is required to maintain the transformed phenotype by LOF experiments, eg. knock down. Do the authors have the possibility to perform such assays in fusion-expressing cell line models? What do they observe?*

Response: We agree that such experiments would be of interest, but there are no patient-derived cell lines representing this new entity yet that would allow us to perform genetic loss-of-function studies. We did, however, have the opportunity to test the sensitivity of tumor cells from patient TFCP2-HD-4, which could be kept in culture to conduct one experiment, to crizotinib, ceritinib, and alectinib (**Fig. 3e**). Since this patient's tumor expressed both ALK-ST2 and ALK-ST3, it was not possible to draw conclusions about variant-specific sensitivities.

To gain insight by other means into the ALK variants' capacity to maintain a transformed phenotype, we have performed an additional experiment and included the results in the revised manuscript (**Fig. 3d**, lines 295-301 [results] and 505-511 [discussion]). Briefly, we assessed the sensitivity of MCF10A cells stably expressing the three new ALK variants, wildtype ALK (ALK-WT), TPM3-ALK as positive control, or empty vector (EV) to crizotinib, ceritinib, and alectinib. Since parental MCF10A cells require EGF, maintenance of a transformed phenotype by ALK variants can be read out by cytokine independence. This experiment yielded the following results:

1. The known oncogenic TPM3-ALK fusion is equally sensitive to all three inhibitors and is the most responsive variant.
2. The three new oncogenic ALK variants differ in their sensitivity to ALK inhibition, with ALK-ST1 being almost as responsive as TPM3-ALK, ALK-ST2 having intermediate sensitivity, and ALK-ST3 being nearly unresponsive.
3. The three inhibitors show differential efficacy against the novel ALK variants, which are best inhibited by ceritinib, followed by crizotinib and alectinib. Of note, we had observed precisely the same differential efficacy in the tumor cells from patient TFCP2-HD-4, which were highly sensitive to ceritinib, showed moderate response to crizotinib, and were resistant to alectinib (**Fig. 3d**).

These new data provide a sensitivity profile of the ALK variants to three clinically approved ALK inhibitors compared to TPM3-ALK, the primary driver of inflammatory myofibroblastic tumor (Lawrence et al. 2000), which confers sensitivity to ALK inhibition (Casanova et al. 2020).

References:

Casanova et al. *Inflammatory myofibroblastic tumor: The experience of the European pediatric Soft Tissue Sarcoma Study Group (EpSSG)*. *Eur J Cancer* 127:123–129 (2020)

Lawrence et al. *TPM3-ALK and TPM4-ALK oncogenes in inflammatory myofibroblastic tumors. Am J Pathol* 157:377–384 (2000)

3. *It remains unclear which cell lines were derived from which patient and how well their response to ALK inhibitors or genotoxic treatment in vitro is associated with the response observed in the patients these cells were derived from. This is an essential piece of information that would help the reader interpret the results of these experiments.*

Response: As described above, we only had the opportunity to test the drug sensitivity of tumor cells from patient TFCP2-HD-4 in short-term culture (**Fig. 3e**). Stable cell lines representing the newly discovered entity of TFCP2-rearranged sarcomas have not yet been established.

Patient TFCP2-HD-4 received the following treatment (**Supplementary Table 1**): Primary surgery, surgery and adjuvant radiotherapy at first relapse after nine months, surgery at second relapse after 12 months, surgery at third relapse after four months (this tumor sample was used for *ex vivo* drug sensitivity testing), doxorubicin and ifosfamide, pazopanib at progression after three months, radiotherapy at symptomatic locoregional progression after one month, complicated by recurrent pleural effusion necessitating repeated pleurocenteses. After completion of radiotherapy, the patient was transferred to a local hospital for further treatment of the pleural effusion, where he was started on ceritinib, but his condition rapidly deteriorated, and he died one month later without radiologic response assessment.

As can be seen from this history, *ex vivo* response to ALK inhibitors was tested in tumor cells obtained after multiple surgeries and adjuvant radiotherapy before systemic chemotherapy. However, the patient received ceritinib at a very late disease stage and in reduced general condition after receiving aggressive chemotherapy, the multi-kinase inhibitor pazopanib, and radiotherapy, all of which certainly drove tumor evolution. This discrepancy complicates the direct comparison of *ex vivo* and *in vivo* efficacy data and could explain the different responses to ALK inhibition of cultured tumor cells compared with the clinical situation. We have included these considerations in the Discussion section of the manuscript (lines 526-533).

When we obtained this patient's tumor sample, we had not yet generated data on the association between TFCP2 fusions and DNA repair defects and, therefore, did not test DNA HRD-directed agents such as cisplatin.

4. *In general, it would be important to understand if high ALK expression of ALK variant expression drives high signaling in these cells. Unfortunately, no evidence is provided about the activation of ALK signaling.*

Response: This is an important point that we addressed with further experimentation. We determined the activity of signaling pathways downstream of ALK in MCF10A cells stably transduced with EV, ALK-WT, and the three oncogenic ALK variants by immunoblotting (new **Supplementary Fig. 3c** and lines 278-280). Specifically, we analyzed the MEK-ERK, PI3K-AKT, and JAK3-STAT3 cascades using phosphorylation-specific antibodies. We found increased phosphorylation of ERK in cells expressing ALK-ST1, ALK-ST2, and ALK-ST3 compared with EV and ALK-WT, with ALK-ST2 having the strongest effect. AKT was highly phosphorylated at Ser473 and slightly at Thr308 only in cells harboring ALK-ST2, and STAT3 Tyr705 showed no increased phosphorylation in any ALK variant compared with EV and ALK-

WT (**Supplementary Fig. 3c**). These data show that ALK-ST2 most strongly activates downstream signaling, i.e., the MEK-ERK and PI3K-AKT pathways, whereas ALK-ST1 and ALK-ST3 activated only the MEK-ERK pathway, and none of the variants led to STAT3 phosphorylation.

In addition, we performed immunohistochemistry on patient tumors using phosphorylation-specific antibodies. These analyses showed consistent phosphorylation of ERK1/2 (Thr202/Tyr204) and PKC δ/θ (Ser643/676), patchy expression of phosphorylated STAT3 (Tyr705), and heterogeneous weak positivity of phosphorylated AKT (Ser473), which is known to be difficult to detect in formalin-fixed and paraffin-embedded tissue specimens. We added a representative image of these stainings from patient TFCP2-HD-4 to the Supplementary Information (**Supplementary Fig. 3d**), and the quantification of phosphorylation signals for all investigated tumors was added to **Supplementary Table 1**. The Results section has been amended accordingly (lines 280-283).

5. The authors demonstrate that the fusion leads to defects in myogenic differentiation, but do not provide mechanistic insights of how this happens. The ACT seq, RNAseq and ChIP seq should be useful to address this question. Can the authors identify pathways that are related to disrupted muscle differentiation there that are directly dysregulated by the fusion through their binding to genes that would promote de-differentiation?

Response: To address this question, we reanalyzed the RNA-seq and ACT-seq data and observed that FUS-TFCP2 and TFCP2 induce molecular programs indicating (partial) myogenic differentiation in the cell line models used.

Specifically, we analyzed the RNA-seq data obtained in SCP-1 cells transduced with FUS-TFCP2 or EV using DAVID (<https://david.ncicrf.gov>) to identify deregulated gene expression programs. In FUS-TFCP2-expressing SCP-1 cells, 283 gene sets were significantly overrepresented (Benjamini-Hochberg [BH]-adjusted p-value <0.05), 12 of which were associated with muscle-related processes defined by the words “muscle” or “myogenesis” in their names, e.g., the gene ontology (GO) terms *MUSCLE_SYSTEM_PROCESS* (BH-adjusted p-value 0.00041), *MUSCLE_TISSUE_DEVELOPMENT* (BH-adjusted p-value 0.01808), or the hallmark gene set *MYOGENESIS* (BH-adjusted p-value 0.04858).

Next, we performed enrichment analysis on H3K27ac ACT-seq peaks using clusterProfiler 3.6.0 (<https://bioconductor.org/packages/release/bioc/html/clusterProfiler.html>). To this end, we determined the thirty peaks with the highest and lowest log₂(fold change) between MCF10A cells transduced with TFCP2 versus EV. This analysis identified a significant overrepresentation of H3K27ac ACT-seq peaks near genes related to the hallmark gene set *MYOGENESIS* (p-value 0.002163, false discovery rate [FDR] 0.09897). When we performed the same analysis with FUS-TFCP2- versus EV-transduced MCF10A cells, we did not see formally significant enrichment of peaks after correcting for multiple testing. However, of all 22,596 gene sets tested, *GO_NEGATIVE_REGULATION_OF_CARDIOCYTE_DIFFERENTIATION* was on the third rank when sorting by p-value, and *GO_CARDIAC_MUSCLE_CELL_FATE_COMMITMENT* was on the second rank when sorting by enrichment ratio, again suggesting that FUS-TFCP2 is involved in transcriptional activation of muscle-related genes.

In line with these findings, *IGFBP5* and *PTH1R*, which are involved in muscle cell proliferation and differentiation (Duan & Allard 2020, Kimura & Yoshioka 2014), were among the 16 genes

deregulated in both MCF10A and SCP-1 cells expressing EWSR1-TFCP2 or FUS-TFCP2 (**Fig. 5c, Supplementary Fig. 5e**). These genes were also more highly expressed in TFCP2-rearranged patient samples than in 90% of tumors analyzed in the MASTER program (**Fig. 5d**). Finally, *IGFBP5* and *PTH1R* expression was significantly upregulated in FUS/EWSR1-TFCP2 tumors compared to all other RMS types (**Supplementary Fig. 2h**). Please see the Discussion section of the revised manuscript for more details (lines 564-582).

References:

Duan & Allard. *Insulin-like growth factor binding protein-5 in physiology and disease. Front Endocrinol* 11:100 (2020)

Kimura & Yoshioka. *Parathyroid hormone and parathyroid hormone type-1 receptor accelerate myocyte differentiation. Sci Rep* 4:5066 (2014)

6. *The ACT-seq experiments do not show very strong signals/ peaks. I am not an expert in this method, but I am not sure if the peaks shown in the main figure really support binding to ALK for example. Could the authors quantify the peak strength and/or rank the peaks and show where ALK and TERT rank compare to other peaks? Also, the ChIPseq data for some active chromatin marks eg H3K27Ac does not seem to change after fusion expression. It would be ideal to compare the binding of the fusion to the degree of chromatin activation. Does the fusion really increase chromatin activation?*

Response: These are valid points. To address the first question, we called differential peaks in EV cells versus cells expressing HA-TFCP2 or HA-FUS-TFCP2. In total, 189 and 94 significant differential peaks were identified for HA-TFCP2 and HA-FUS-TFCP2, respectively. For TFCP2 and FUS-TFCP2, significant peaks were detected in regions annotated to *TERT* and *ALK*, respectively. These peaks were highly significant and ranked well below a FDR of 10%, as illustrated in **Response Fig. 4**, demonstrating that the TFCP2 and FUS-TFCP2 binding events at the *TERT* and *ALK* loci are unlikely to be false positives. The validity of our ACT-seq data is further underlined by the analysis provided in **Supplementary Fig. 5g**, which demonstrates that the motifs enriched in peaks found in HA-TFCP2 and HA-FUS-TFCP2 cells closely resemble the known Tfcp2l1 consensus motif (<http://homer.ucsd.edu/homer/motif/motifDatabase.html>).

Response Fig. 4: p-values (y-axis) ranked in increasing order (x-axis) for all differential peaks called in EV- versus HA-TFCP2-transduced MCF10A cells (left) or in EV- versus HA-FUS-TFCP2-transduced MCF10A cells (right). Peaks annotated to *TERT* and *ALK* are highlighted in red. Peaks with a FDR <10% are to the left of the dotted line.

To address the second question, we compared H3K27ac signal intensities in regions showing differential binding of TFCP2 or FUS-TFCP2 compared to EV. However, similar to what the reviewer observed at the *ALK* locus, no significant differences in global H3K27ac deposition at regions bound by TFCP2 (p-value 0.975, Wilcoxon test) or FUS-TFCP2 (p-value 0.777, Wilcoxon test) were found (**Response Fig. 5**). In summary, our experimental data do not provide evidence for differential H3K27ac deposition at loci bound by FUS-TFCP2, suggesting that FUS-TFCP2 binding might activate target genes by alternative mechanisms. This is supported by the fact that the TFCP2 binding site upstream of *ALK* exhibits H3K27ac deposition in EV cells, which remains unchanged upon FUS-TFCP2 expression although *ALK* mRNA and protein are induced (**Fig. 5a**; **Supplementary Fig. 4a, b**; **Supplementary Fig. 5a, b**).

Response Fig. 5: Average normalized H3K27ac signal at TFCP2 (left) and FUS-TFCP2 (right) peaks. H3K27ac profiles were plotted for EV controls (black) and cells expressing HA-TFCP2 (red, left) or HA-FUS-TFCP2 (red, right).

7. It remains unclear how the fusion influences DNA damage repair. It does seem that expression of the fusion is sufficient to increase DNA damage, but it is unclear what causes this DNA damage, eg. could it be fusion-induced replication stress as observed for EWS-FLI1 in Ewing's sarcomas?

Response: We agree. The link between the expression of TFCP2 fusions and defective DNA damage is intriguing but functionally not understood. As suggested by the reviewer, fusion gene-induced replication stress, as observed in Ewing sarcoma (Gorthi et al. 2018), might be involved. We therefore investigated the ATR-mediated replication stress response in MCF10A cells expressing EV, FUS-TFCP2, FUS, or TFCP2 by immunoblotting. As shown in **Response Fig. 5**, we did not observe increased phosphorylation of the central replication stress response kinase ATR or activation of the ATR-dependent effectors CHK1 and RPA32.

Response Fig. 5: Immunoblot analysis of MCF10A cells transduced with EV or HA-tagged FUS-TFCP2, FUS, or TFCP2 for replication stress-specific markers using the same antibodies as in Gorthi et al. 2018. Two independent biological replicates are shown.

Another mechanism by which FUS-TFCP2 could compromise DNA repair is the loss of wildtype FUS, which we briefly discussed in the manuscript (lines 587-592). FUS localizes to sites of DNA damage and is required for efficient repair of DNA double-strand breaks by homologous recombination or non-homologous end joining (Mastrocol et al. 2013, Sama et al. 2014), functions that require C-terminal domains that are lost in the TFCP2 fusions. In addition, we observed downregulation of wildtype FUS after exogenous expression of FUS-TFCP2 (**Response Fig. 6**), suggesting that the unaffected FUS allele may not be able to compensate for the FUS portion lost in the rearranged allele but may even be suppressed. Similar considerations could apply to EWSR1-TFCP2, as functions in DNA damage response have also been identified for EWSR1 (Lee et al. 2020, Paronetto et al. 2011). This observation is now mentioned in the Discussion section of the manuscript (lines 588-590).

Author Response Fig. 6: Immunoblot analysis of MCF10A (left) and LHCN-M2 (right) cells transduced with EV or HA-tagged FUS, TFCP2, or FUS-TFCP2 with an anti-FUS antibody (Sigma-Aldrich, SAB2108528). Endogenous FUS was suppressed in both cell lines after expression of FUS-TFCP2. R1, R2, and R3 indicate biological replicates.

In addition, TERT is involved in DNA repair independently of its canonical immortalization function (Pestana et al. 2017). For example, as outlined in the Discussion section of the manuscript (lines 592-596), cells with persistent TERT loss exhibit altered chromatin structure, impaired activation of the DNA damage response, enhanced radiosensitivity, and increased numbers of chromosome fragments upon irradiation (Masutomi et al. 2005). Thus, binding of TFCP2 fusions to the *TERT* locus upstream of exon 3 could cause expression of the non-functional Δ ex1-2 variant, which could phenocopy TERT loss and hence impair DNA repair.

Overall, there are many possibilities how TFCP2 fusions could affect DNA repair. We believe that deciphering the underlying complex mechanisms will require extensive experimentation, which should be the subject of future studies and is beyond the scope of this manuscript.

References:

Gorthi et al. *EWS-FLI1 increases transcription to cause R-loops and block BRCA1 repair in Ewing sarcoma. Nature* 555:387–391 (2018)

Lee et al. *Ewing sarcoma protein promotes dissociation of poly(ADP-ribose) polymerase 1 from chromatin. EMBO Rep* 21:e48676 (2020)

Mastrocol et al. *The RNA-binding protein fused in sarcoma (FUS) functions downstream of poly(ADP-ribose) polymerase (PARP) in response to DNA damage. J Biological Chem* 288: 24731–24741 (2013)

Paronetto et al. *The Ewing sarcoma protein regulates DNA damage-induced alternative splicing. Mol Cell* 43:353–368 (2011)

Sama et al. *Functions of FUS/TLS from DNA repair to stress response: implications for ALS. Asn Neuro* 6:1759091414544472 (2014)

Reviewer #3 (Remarks to the Author): Expert in protein structure prediction and structural biology

I was asked to review especially the structural modelling part of the work of Schöpf et al. thus I limit my assessment exclusively on this part. For the use the authors are doing of the structural model produced for the human ALK protein I think the modelling is well done and sufficient to interpret their data, as they limit their analysis only to structured domains whose homology is well captured by HHpred. A more comprehensive model is provided also at the AlphaFold database (<https://alphafold.ebi.ac.uk/entry/Q9UM73>) which the authors can also use and/or compare with their model. Note that in the AF model, while the single domains are well predicted, likely with similar accuracy as HHpred, their reciprocal location is uncertain (see the large expected position error), thus the comparative analysis might have some meaning only for the single domains.

Response: Thank you for this great suggestion. The AlphaFold structures were not available when we initially did the structural modeling. We did inspect the model for the human ALK protein in the AlphaFold database as soon as it was published. However, we noted some major issues with the structure, such as the placement of the domains in three-dimensional space. For instance, the AlphaFold structure places the kinase domain close to the extracellular domains, which is problematic. This is also understandable, as AlphaFold does not account for transmembrane proteins. We have now added the following text to the supplementary methods (page 7 in the Supplementary Information file) to discuss this aspect:

“The complete structure for human ALK-WT has been predicted by AlphaFold2 (Tunyasuvunakool et al. 2021) and is publicly available (<https://alphafold.ebi.ac.uk/entry/Q9UM73>). While the single domain predictions seemed to be reasonable, the overall structure was problematic with respect to the placement of the domains in three-dimensional space. For instance, the kinase domain is closely placed with the extracellular domains, which is not a valid topology. Indeed, AlphaFold does not account for transmembrane proteins, and hence the predicted structures for such proteins can be topologically inaccurate. Thus, we used the HHpred web server”

The reviewer is also correct in pointing out that the single domain predictions by AlphaFold themselves could be used for modeling. Thus, we analyzed the structural similarity between the MAM-1, LDL receptor, and MAM-2 domains of our modeled structure and the AlphaFold model using TM-align (Zhang & Skolnick 2005). We found that the TM-scores were 0.62, 0.64, and 0.70, respectively, which indicates a high structural similarity akin to belonging to the same protein fold. The structures for the remaining domains, extracellular region and kinase, were taken from published structures of these domains in the Protein Data Bank. Thus, they should be the most accurate representation. We hope the reviewer agrees that our model is good enough compared with the single domain predictions by AlphaFold, and hence, a change to the figure is not quite necessary.

References:

Tunyasuvunakool et al.. Highly accurate protein structure prediction for the human proteome. Nature 596:590–596 (2021)

Zhang & Skolnick. TM-align: a protein structure alignment algorithm based on the TM-score. Nucleic Acids Res 33:2302–2309 (2005)

REVIEWERS' COMMENTS

Reviewer #1 (Remarks to the Author):

I have reviewed the authors' responses to my prior critiques, and am satisfied with their careful, detailed, and thorough responses.

I defer to the journal editors on whether or not it is acceptable to refer to "data not shown," as stated in the revised text line 268 regarding their confirmation of ALK-ST4 by proteomics; or whether the authors should include their additional proteomics analysis in Supplementary Data.

Reviewer #2 (Remarks to the Author):

All my comments were addressed to a great extent using new experimental data. I appreciate the authors work to address all of these comments and commend them on their work. I look forward to seeing this work in print.

Reviewer #3 (Remarks to the Author):

The authors responded very well to all my points - a residual point in the text they add regarding the AF model: they can also use the high PAE value to quantify what they expressed in words

Reviewer #4 (Remarks to the Author): Technical reviewer with expertise in ACT-seq and epigenetics

I was asked to comment on the ACT-seq data. Reviewer 2 noted that the ACT-seq signals/peaks are not very strong, which I agree. It appears to me that some of the ACT-seq samples may have a very limited number of sequencing reads in the libraries, which might be partially responsible for the low signal/peak levels in the genome browser tracks. Furthermore, the authors identified only 189 and 94 significant peaks for HA-TFCP2 or HA-FUS-TFCP2, respectively, as compared to the EV control samples. These numbers are pretty small for TF binding sites. This might also be caused by the low sequencing depth of

the libraries. However, the authors showed that the consensus motifs identified from these binding sites are almost identical with that of the mouse ortholog Tcfcp2l1 based on ChIP-seq data, suggesting that these peaks are pretty reliable, although the numbers are small. Thus, I feel the conclusion of HA-TFCP2 or HA-FUS-TFCP2's binding to the ALK and TERT genes is solid. However, there might be many false negative sites in the genome due to the low sequencing reads in the libraries, which the authors need to mention.

Regarding whether chromatin is activated or associated with higher levels of H3K27ac at the regions bound by HA-TFCP2 or HA-FUS-TFCP2, I feel the authors need to be more cautious when concluding that there is no chromatin activation, because the background signals of the H3K27ac ACT-seq data look pretty high and there could be many false negative sites for HA-TFCP2 or HA-FUS-TFCP2 binding.

Reviewer #5 (Remarks to the Author): Technical reviewer with expertise in proteomics

The article provides a comprehensive multi-omics characterization of sarcomas with FUS-TFCP2 or EWSR1-TFCP2 fusions. Through an in-depth analysis, the authors gain insights into the mechanisms underlying these fusion events. As I have been specifically asked to review the proteomics section of the article, I will therefore restrict the review to this part.

Overall, the proteomics experiment design, DIA data generation, and DIA analysis appear to be standard to me. More than 10 supporting peptides were identified for ALK-ST4 which enhances the reliability of this identification and quantification. Here are several comments that may help the authors in enhancing the interpretation and reproducibility of their proteomics data.

1. The protein database and DIA-NN results should be submitted to PRIDE along with your raw and mzML files.
2. The sections of "Sample Processing Protocol" and "Data Processing Protocol" can be incorporated into the method part.
3. The following parameters much be specified in the "Data Processing Protocol" section: Protease, deep learning-based spectra, RTs, and IMs prediction, peptide length, modifications, and mass accuracy.
4. Even though these are DIA data, including several spectrum-peptide matching figures may help to illustrate the identified peptides.

POINT-BY-POINT RESPONSE TO THE REVIEWERS' COMMENTS

Reviewer #1 (Remarks to the Author):

I have reviewed the authors' responses to my prior critiques, and am satisfied with their careful, detailed, and thorough responses.

I defer to the journal editors on whether or not it is acceptable to refer to "data not shown," as stated in the revised text line 268 regarding their confirmation of ALK-ST4 by proteomics; or whether the authors should include their additional proteomics analysis in Supplementary Data.

Response: We agree and moved the validation of ALK-ST4 protein expression by mass spectrometry to the main manuscript. The respective figure is now displayed in Supplementary Figure 3 and the procedure is explained in the Supplementary Methods.

Reviewer #2 (Remarks to the Author):

All my comments were addressed to a great extent using new experimental data. I appreciate the authors work to address all of these comments and commend them on their work. I look forward to seeing this work in print.

Reviewer #3 (Remarks to the Author):

The authors responded very well to all my points - a residual point in the text they add regarding the AF model: they can also use the high PAE value to quantify what they expressed in words.

Response: This is a very good point, and we have modified the structural predictions section of the Supplementary Methods accordingly. Specifically, we replaced the sentence "Indeed, AlphaFold does not account for transmembrane proteins and hence the predicted structures for such proteins can be topologically inaccurate" with the following text: "Indeed, this was also evident in the metric used by AlphaFold that assigns confidence to the relative positions of individual domains, namely Predicted aligned error (PAE). The PAE values for the kinase domain versus the extracellular MAM-1, LDL receptor, and MAM-2 domains were consistently >25 indicating low confidence in their relative positions (see <https://alphafold.ebi.ac.uk/entry/Q9UM73> and PAE values for scored residues 200–620 versus aligned residues 1.1k–1.4k)."

Reviewer #4 (Remarks to the Author): Technical reviewer with expertise in ACT-seq and epigenetics

I was asked to comment on the ACT-seq data. Reviewer 2 noted that the ACT-seq signals/peaks are not very strong, which I agree. It appears to me that some of the ACT-seq samples may have a very limited number of sequencing reads in the libraries, which might be

partially responsible for the low signal/peak levels in the genome browser tracks. Furthermore, the authors identified only 189 and 94 significant peaks for HA-TFCP2 or HA-FUS-TFCP2, respectively, as compared to the EV control samples. These numbers are pretty small for TF binding sites. This might also be caused by the low sequencing depth of the libraries. However, the authors showed that the consensus motifs identified from these binding sites are almost identical with that of the mouse ortholog Tcfcp2l1 based on ChIP-seq data, suggesting that these peaks are pretty reliable, although the numbers are small. Thus, I feel the conclusion of HA-TFCP2 or HA-FUS-TFCP2's binding to the ALK and TERT genes is solid. However, there might be many false negative sites in the genome due to the low sequencing reads in the libraries, which the authors need to mention.

Regarding whether chromatin is activated or associated with higher levels of H3K27ac at the regions bound by HA-TFCP2 or HA-FUS-TFCP2, I feel the authors need to be more cautious when concluding that there is no chromatin activation, because the background signals of the H3K27ac ACT-seq data look pretty high and there could be many false negative sites for HA-TFCP2 or HA-FUS-TFCP2 binding.

Response: Thank you for carefully reviewing the ACT-seq experiment and the very helpful feedback. According to the reviewer's suggestion, we added the underlined text in the results section: Despite the low complexity of the libraries, which can lead to false negative sites in the genome, we were able to predict the transcription factor binding motif of TFCP2 from the ACT-seq data, which consisted of 15 nucleotides and was nearly identical to that of the mouse ortholog Tcfcp2l1 inferred from chromatin immunoprecipitation sequencing data in the HOMER database 44 (Supplementary Fig. 5f). The predicted binding motif of FUS-TFCP2 contained the same sequence but with an additional repeat of nine nucleotides at the 3' end (Supplementary Fig. 5f). This suggested that the detected peaks are reliable, although their numbers might be underestimated. Inspection of specific gene loci revealed that TFCP2 and FUS-TFCP2 bound to the TFCP2 binding motif CAGCCCTGTCCAGTCCAGTT in a genomic region 45 kilobases (kb) upstream of the ALK transcriptional start site and 9 kb upstream of the unannotated ALK exons -3, -2, and -1, where the transcriptional activation mark H3K27ac appeared to be enriched (Fig. 5e; Supplementary Fig. 5g).

Reviewer #5 (Remarks to the Author): Technical reviewer with expertise in proteomics

The article provides a comprehensive multi-omics characterization of sarcomas with FUS-TFCP2 or EWSR1-TFCP2 fusions. Through an in-depth analysis, the authors gain insights into the mechanisms underlying these fusion events. As I have been specifically asked to review the proteomics section of the article, I will therefore restrict the review to this part.

Response: Thank you for carefully reviewing our manuscript and the constructive feedback on the proteomics experiment.

Overall, the proteomics experiment design, DIA data generation, and DIA analysis appear to be standard to me. More than 10 supporting peptides were identified for ALK-ST4 which enhances the reliability of this identification and quantification. Here are several comments that may help the authors in enhancing the interpretation and reproducibility of their proteomics data.

1. The protein database and DIA-NN results should be submitted to PRIDE along with your raw and mzML files.

Response: The protein database and the DIA-NN results were uploaded to PRIDE and are now publicly accessible under the identifier PXD045522. Specifically, the PRIDE upload contains .raw, mzML, the fasta file used, and the complete search result output from DIA-NN.

2. The sections of “Sample Processing Protocol” and “Data Processing Protocol” can be incorporated into the method part.

Response: We moved the proteomics results from the previous response letter to Supplementary Figure 3 and now describe the entire mass spectrometry procedure, including sample and data processing, in the Supplementary Methods.

3. The following parameters must be specified in the “Data Processing Protocol” section: Protease, deep learning-based spectra, RTs, and IMs prediction, peptide length, modifications, and mass accuracy.

Response: As suggested, we included these parameters in the new Supplementary Methods section.

4. Even though these are DIA data, including several spectrum-peptide matching figures may help to illustrate the identified peptides.

Response: We thank the reviewer for this suggestion and agree that such a depiction would be very informative. While the DIA-NN software is an outstanding advance for the analysis of DIA data, one limitation is the lack of a “spectrum viewer” to generate the requested spectrum-peptide match figures. As an alternative, we managed to display for the ALK-WT/ALK-ST4 precursors/peptides the extracted ion chromatograms of their MS2 fragments used in the quantification by DIA-NN. This information is included as a representative image in Supplementary Figure 3. Furthermore, the Source Data file contains all output tables from DIA-NN. Although not precisely aligning with the reviewer's initial suggestion, we hope that the newly included data are considered sufficient.